# Tunable translation-level CRISPR interference by dCas13 and engineered gRNA in bacteria

Giho Kim[1], Ho Joon Kim[1], Keonwoo Kim [1], Hyeon Jin Kim[2], Jina Yang [3] & Sang Woo Seo [1,2,4,5,6] ✉

Although CRISPR-dCas13, the RNA-guided RNA-binding protein, was recently exploited as a translation-level gene expression modulator, it has still been difficult to precisely control the level due to the lack of detailed characterization. Here, we develop a synthetic tunable translation-level CRISPR interference (Tl-CRISPRi) system based on the engineered guide RNAs that enable precise and predictable down-regulation of mRNA translation. First, we optimize the Tl-CRISPRi system for specific and multiplexed repression of genes at the translation level. We also show that the Tl-CRISPRi system is more suitable for independently regulating each gene in a polycistronic operon than the transcription-level CRISPRi (Tx-CRISPRi) system. We further engineer the handle structure of guide RNA for tunable and predictable repression of various genes in *Escherichia coli* and *Vibrio natriegens*. This tunable Tl-CRISPRi system is applied to increase the production of 3-hydroxypropionic acid (3-HP) by 14.2-fold via redirecting the metabolic flux, indicating the usefulness of this system for the flux optimization in the microbial cell factories based on the RNA-targeting machinery.

Developing synthetic gene expression modulators such as CRISPR-Cas system and synthetic sRNA enabled high-throughput screening of genes to systematically understand the microbial system and engineer the metabolic pathways of microorganisms[1–5]. Catalytically dead Cas9 (dCas9) or dead Cas12 (dCas12) can specifically bind to target sequences in double-stranded DNA (dsDNA), block the transcription of RNAP, and silence gene expression[6,7]. Synthetic sRNA has been harnessed to suppress gene expression at the translation level since it can efficiently bind to the mRNA together with the Hfq protein[8].

The Type VI CRISPR-Cas nuclease, called Cas13, is different from Cas9 or Cas12 in that it can specifically target and cleave single-stranded RNA[9,10]. Unlike Cas9 and Cas12, which require strict protospacer-adjacent motif sequences for targeting, Cas13 requires minimal or no protospacer flanking sequences, allowing flexible

targeting of mRNA[11]. Unfortunately, when the nuclease-active state of Cas13 binds to the target RNA and forms the ternary ribonucleoprotein (RNP) complex, it causes non-specific degradation of adjacent RNA, thereby significantly limiting cell growth[9]. This phenomenon has confounded the utilization of Cas13 as a specific gene knockdown tool in bacteria. Thus, the nuclease-inactive state of Cas13, called dead Cas13 (dCas13), has been repurposed to repress the gene expression at the translation level without causing growth retardation by sequestering the translation initiation region of mRNA[12]. Also, gene activation at the translation level was achieved by fusing the translation initiation factor to the dCas13[13]. However, the dCas13-based gene knockdown or activation was only tested for a simple ON/OFF regulation of a monocistronic gene. The characteristics of dCas13 as a synthetic gene expression modulator, such as the specificity or the multiplexity, are

[1]School of Chemical and Biological Engineering, Seoul National University, Seoul, Republic of Korea. [2]Interdisciplinary Program in Bioengineering, Seoul National University, Seoul, South Korea. [3]Department of Chemical Engineering, Jeju National University, Jeju-si, South Korea. [4]Institute of Chemical Processes, Seoul National University, Seoul, South Korea. [5]Bio-MAX Institute, Seoul National University, Seoul, South Korea. [6]Institute of Bio Engineering, Seoul National University, Seoul, South Korea. ✉e-mail: swseo@snu.ac.kr

not yet investigated in detail. The lack of a fine-tuning strategy that can moderately regulate the gene expression by the dCas13 has confined its usage in bacterial cells compared to other well-characterized synthetic gene regulation systems.

In this study, we develop the dCas13-based tunable Tl-CRISPRi system that can precisely tune the translation level of diverse genes in two different bacteria. We apply the Tl-CRISPRi to the reporter genes in the chromosome or plasmid, revealing its titratable, specific, and multiplexed gene knockdown capability. Comparing the Tl-CRISPRi with transcription-level CRISPRi (Tx-CRISPRi) under regulating the expression of a polycistronic operon, we highlight the distinctiveness of the Tl-CRISPRi system in that it can independently down-regulate each gene in the synthetic and native operons. Furthermore, we develop a strategy to fine-tune the expression level of a target gene by modifying the sequence and structure of the guide RNA handle. The library of attenuated guide RNAs allows uniformly distributed target gene expression levels ranging from 2.6% to 86.3% and predictable expression tuning across other genes. We verify that these attenuated guide RNAs robustly work in both *Escherichia coli* and *Vibrio natriegens*, suggesting their expandability toward the other bacterial strains. Finally, we adopt this tunable Tl-CRISPRi system to redirect the metabolic flux for improving the bioproduction of 3-hydroxypropionic acid (3-HP). We repress 48 target genes across metabolic pathways to identify the effective knockdown targets. We utilize attenuated guide RNAs to balance the bacterial growth and bioproduction, resulting in a 14.2-fold enhancement of 3-HP titer. Our tunable Tl-CRISPRi system has the potential to be a robust and predictable translation-level gene regulation system in bacteria.

## Results

### Design and characterization of the dCas13-based Tl-CRISPRi system

Several previous studies have discovered hundreds of Cas13 orthologs that can bind to target RNAs across a wide range of bacterial species with the help of genome mining and classified into four subtypes (Cas13a, Cas13b, Cas13c, and Cas13d) depending on the existence of accessory proteins or the primary amino acid sequences of the Cas13 effectors[14–17]. We selected LbuCas13a, BzCas13b, PbCas13b, and RfxCas13d, of which several biochemical properties and the critical catalytic residues involved in non-specific nuclease activity are already well characterized[15,18,19]. We sought to compare the efficiency of translation repression when these orthologs became catalytically inactive. The catalytic residues were mutated to eliminate non-specific RNA cleavage activity, generating the dCas13 proteins (Supplementary Fig. 1a). In our experimental design, the expression of dCas13 protein was induced by anhydrotetracycline (aTc), and the guide RNA targeting the 5′-untranslated region (5′-UTR) of mCherry mRNA was constitutively expressed in *E. coli* K-12 MG1655 harboring the expression cassette of mCherry (Fig. 1a). While three dCas13 orthologs did not show significant gene knockdown upon the addition of aTc, the catalytically dead version of RfxCas13d (dRfxCas13d) decreased the expression level of the reporter gene to 2.8% compared to the non-targeted control (Supplementary Fig. 1b). The bacterial growth was not significantly perturbed upon the mRNA knockdown of dRfxCas13d (Supplementary Fig. 1c, d), which exhibits robustness as a synthetic gene expression modulator. When we investigated the expression of each dCas13 ortholog by western blotting and coomassie blue staining, only the dRfxCas13d was clearly detected in both the total and soluble fraction of bacterial cells (Supplementary Fig. 1e, f). This result emphasizes the appropriate choice of dCas13 ortholog to be expressed in the functional form for efficient gene knockdown. Therefore, we chose the dRfxCas13d and its cognate guide RNA to construct the synthetic Tl-CRISPRi system.

We first examined whether the binding of dCas13 indeed mediated the knockdown of targeted mRNA since only the formation of

RNA duplex could knock down the mRNA as in the case of antisense RNA[20–22]. We tested various concentrations of aTc, thereby differing the expression level of dCas13, to evaluate the correlation between the intracellular amount of dCas13 and the target gene expression. Notably, the decrease in mCherry level upon the addition of aTc was more remarkable when the *mCherry*-targeting guide RNA was expressed, compared to the non-target strain (Supplementary Fig. 2). The correlation between the concentration of dCas13 inducer and the target gene expression level suggests that dCas13 is a critical component for efficient gene repression via Tl-CRISPRi, which can efficiently sequester the translation initiation site of target mRNA. This result also shows the possible usefulness of this system for biosensor-associated dynamic regulation as performed in the previous CRISPRi studies[23,24]. Since the knockdown by the Tl-CRISPRi was saturated with more than 100 ng/mL of aTc, and some extent of growth retardation was observed over 200 ng/mL of aTc (Supplementary Fig. 2), possibly due to the metabolic burden from the expression of large Cas protein[25], we determined to induce the expression of dCas13 with 100 ng/mL of aTc for the following experiments.

To validate that Tl-CRISPRi could robustly knockdown genes in the various locations of the chromosome, we prepared *E. coli* strains, each harboring reporter gene expression cassette (*mCherry*, *GFP*, *nanoluciferase*) integrated into different loci of the chromosome. Six spacers (sp1–sp6) spanning from 5′-UTR to the N-terminal region of the coding sequence were designed and applied for the knockdown of each reporter gene. We found that Tl-CRISPRi robustly knocked down each gene regardless of its integrated location (Fig. 1b). However, it should be noted that each spacer's knockdown efficiency was somewhat different. We reasoned that the accessibility toward targeted RNA sequence or local AU and GC distribution, previously revealed as significant factors determining the targeting efficiency of Cas13[26,27], could affect the binding efficiency of the dCas13-guide RNA complex toward the target RNA to some degree. Nevertheless, all the spacers targeting near the translation initiation region of mRNA showed remarkable gene knockdown compared to the non-targeted strain. We applied the sp1 spacer of each reporter gene to guarantee the robust repression of gene expression for the following experiments. We next investigated whether the Tl-CRISPRi could specifically and simultaneously repress multiple gene expressions. We found that two or three reporter genes were successfully knocked down by introducing multiple guide RNA expression cassettes (Fig. 1c, d). This result demonstrates the specificity and the multiplexity of the Tl-CRISPRi system.

Next, we tested whether the Tl-CRISPRi could efficiently inhibit the translation of mRNA transcribed from the plasmid and whether the amount of the target mRNA could affect the knockdown efficiency. Four constitutive promoters with different transcription strengths were adopted to express the mCherry on the plasmid (Supplementary Fig. 3a). We found that the translation of mCherry mRNAs transcribed from the plasmid could also be efficiently repressed by the Tl-CRISPRi (Supplementary Fig. 3b). Additionally, the resulted relative expression of the mCherry by the Tl-CRISPRi did not significantly fluctuate regardless of the transcription rate of the *mCherry* (Supplementary Fig. 3c), showing from 3.8% ($P_{J23100}$) to 6.4% ($P_{J23113}$).

Since optimizing the spacer length could reduce the possibility of forming a secondary structure inside itself and enhance the RNA-targeting efficiency[28], we sought to find the minimum length of the spacer that can retain the knockdown efficiency. To this end, we changed the lengths of complementary sequences to the mCherry mRNA in spacers by introducing mismatches. We found that the expression level of mCherry gradually increased when the length of the spacer-protospacer complementarity was lower than 22 nts (Supplementary Fig. 4a). This result is consistent with the previous study where the gene knockdown efficiency of RfxCas13d was maintained over the 23-nt length of the spacer in the mammalian cell[28,29]. Truncated spacers with a length of 27 nts, 23 nts, and 20 nts were also tested

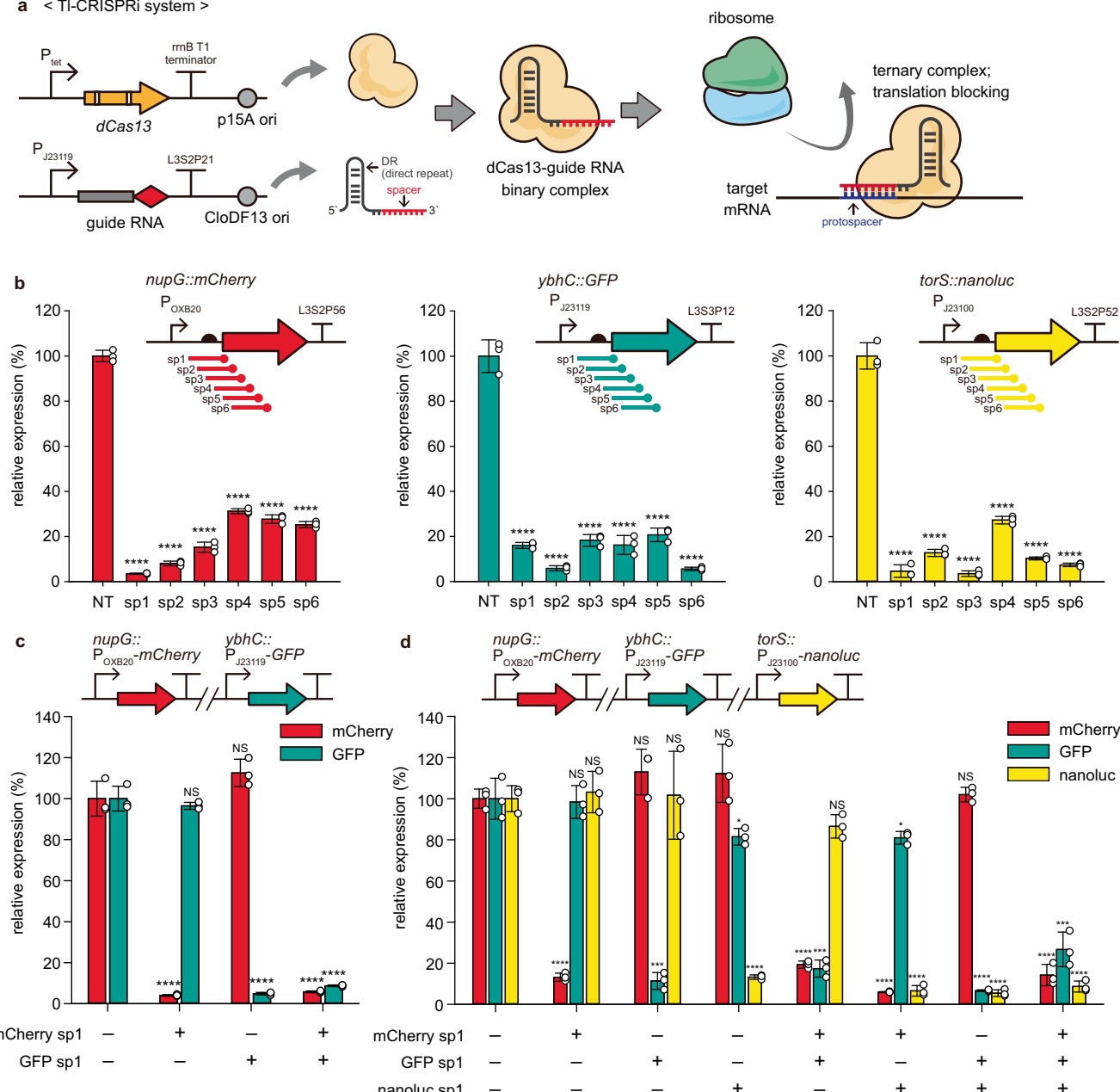

**Fig. 1 | Fundamental characterization of Tl-CRISPRi in the *E. coli* K-12 MG1655.**
**a** A scheme of dCas13-based translation repression. The dCas13 protein and its cognate guide RNA are assembled, forming a RNP binary complex. We redirected it to bind to the translation initiation site of specific mRNA to block the translation of the ribosome and turn off the expression of a target gene. **b** Adopting the Tl-CRISPRi for three different reporter genes (*mCherry*, *GFP*, *nanoluc*) with six different spacers for each gene. NT represents the strain with the non-target guide RNA (RfxgRNA-NT). **c, d** Simultaneous and specific reporter gene knockdown by the Tl-CRISPRi. Each guide RNA with the effective spacer (mCherry sp1, GFP sp1, nanoluc sp1) specifically down-regulated the target gene. Simultaneous gene

knockdown was achieved by introducing multiple guide RNAs. The relative expression (**b**–**d**) was determined based on the specific fluorescence or luminescence level (RFU or RLU/OD$_{600}$) and normalized by setting the value of the NT strain as 100%. The error bar represents the mean ± standard deviation from the biologically independent cell cultures ($n = 3$), and the white dots indicate the actual data points. The *P*-value of each strain's dataset was determined by the two-tailed Student's *t*-test compared to the dataset of the NT strain. The asterisk above the bar indicates the *P*-value. NS not significant; *$P < 0.05$, **$P < 0.01$, ***$P < 0.001$, ****$P < 0.0001$. Source data are provided as a source data file.

for further validation, anchoring the position of the 3′ end of the spacer. Indeed, tight knockdown of the reporter gene was observed when the length of the spacer was reduced to 23 nts, confirming that the minimal size of the spacer should be 23 nts (Supplementary Fig. 4b).

We next investigated whether the amount of targeted mRNA could also be affected by Tl-CRISPRi. Because of the co-localized and coordinated transcription/translation processes in a prokaryotic cell,

suppressing the translation of the mRNA could lead to its degradation and premature transcription termination due to the absence of ribosomes protecting the mRNA[30]. We found that the Tl-CRISPRi decreased the quantity of all three reporter mRNA, although it was less significant than the decrease in protein level (Supplementary Fig. 5). This result suggests that excluding the ribosome by blocking translation initiation via Tl-CRISPRi could moderately decrease the amount of targeted mRNA, even though the dCas13 lacks the function of blocking

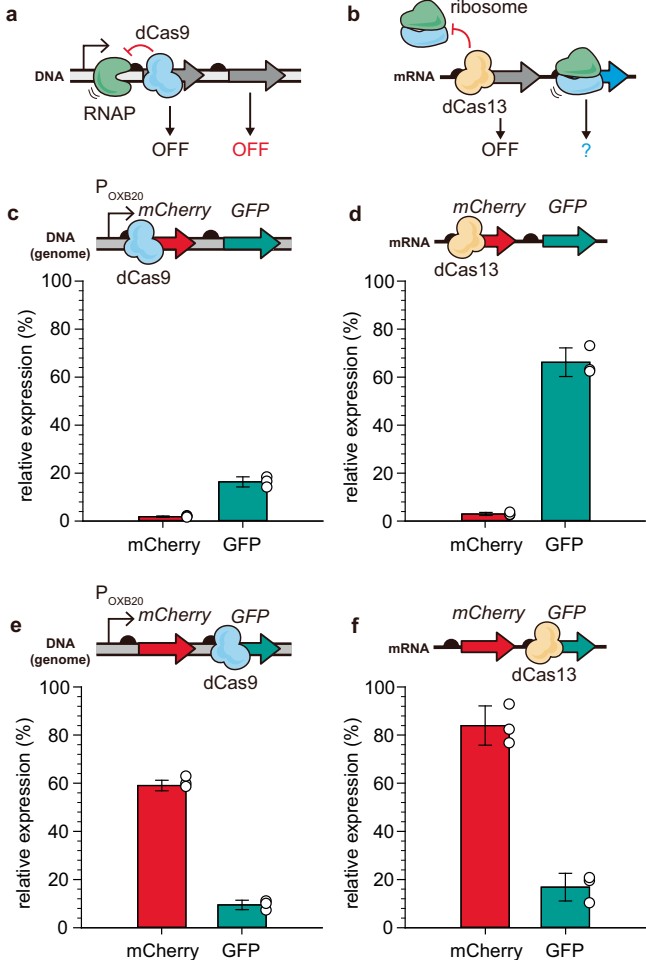

**Fig. 2 | Comparing the Tx-CRISPRi and the Tl-CRISPRi on regulating the polycistronic gene expression (*mCherry-GFP*).** The reporter gene operon consisting of *mCherry* and *GFP* was expressed from the chromosome of *E. coli*. **a**, **b** The expected difference between transcription-level control and translation-level control toward polycistronic gene expression. If the dCas9 blocks the upstream gene in the operon, all downstream genes are simultaneously turned off, called the polar effect. We reasoned that dCas13-based translation-level knockdown on the upstream gene would not intrinsically down-regulate the expression of downstream genes in the operon if each cistron is independently translated from its ribosome binding site. **c**–**f** The relative expression level when each reporter gene was targeted by the Tx- or Tl-CRISPRi. For the Tl-CRISPRi, the spacer mCherry sp1 or GFP sp1 was applied to knock down the expression of mCherry or GFP, respectively. The relative expression was determined based on the specific fluorescence level (RFU/OD$_{600}$) and normalized by setting the value of the NT (non-targeted) strain as 100%. The error bar represents the mean ± standard deviation from the biologically independent cell cultures (*n* = 3), and the white dots indicate the actual data points. Source data are provided as a source data file.

transcription by targeting the double-stranded DNA[9,15] or recruiting endogenous RNases as sRNA-Hfq does[31,32].

## Regulating the polycistronic operon expression via the Tl-CRISPRi

When targeting a polycistronic operon, the conventional dCas9-based Tx-CRISPRi turns off the expression of all cistrons located downstream of a target gene[33], known as the polar effect[34] (Fig. 2a). We assumed that Tl-CRISPRi would independently repress each gene in an operon, unlike the Tx-CRISPRi, since Tl-CRISPRi could be specifically directed to the translation initiation site of each cistron without disturbing the translation of neighboring cistrons in the same transcript (Fig. 2b). To

test this hypothesis, we introduced Tx-CRISPRi or Tl-CRISPRi, both targeting near the N-terminal region of each gene in a polycistronic operon comprised of *mCherry* and *GFP*. When the upstream *mCherry* was targeted, the Tx-CRISPRi led to far reduced expression of the downstream *GFP* (19.6%) compared to that (63.1%) by the Tl-CRISPRi (Fig. 2c, d). When the downstream *GFP* was targeted, the Tx-CRISPRi and the Tl-CRISPRi successfully down-regulated the expression of *GFP* (Fig. 2e, f). This result suggests that the Tl-CRISPRi could be harnessed for a more specific knockdown of each cistron in the operon than the conventional Tx-CRISPRi, which was also validated in tri-cistronic reporter operon comprised of *GFP*, *nanoluciferase*, and *mCherry* in series (Supplementary Fig. 6). However, it should be noted that the Tl-CRISPRi also showed some degree of polar effect. We reasoned that the accelerated mRNA degradation due to the decoupled transcription and translation could be responsible for the polar effect (Supplementary Fig. 5). Since a similar polar effect from translation inhibition was also observed in a previous study utilizing Hfq-sRNA[35], it should be considered that some extent of the polar effect is unavoidable under the translation-level gene knockdown. Nevertheless, the expression of downstream genes was not significantly turned off when Tl-CRISPRi targeted the upstream gene.

Beyond the reporter genes, we next compared the Tl-CRISPRi with Tx-CRISPRi by targeting the native operons of *E. coli*, consisting of upstream non-essential genes and downstream essential genes, to evaluate the influence of polar effect in bacterial growth. In the previous study, Tx-CRISPRi toward non-essential upstream genes in these operons hampered bacterial growth owing to the polar effect[34]. We anticipated that Tl-CRISPRi would less impede cellular growth when targeting these genes due to the diminished polar effect (Fig. 3a). Five endogenous operons were selected for the proof of concept, and the Tx-CRISPRi or the Tl-CRISPRi targeted upstream non-essential genes (Fig. 3b). As expected, introducing the Tx-CRISPRi targeting upstream non-essential genes caused remarkable growth retardation (Fig. 3c). In contrast, strains with the Tl-CRISPRi targeting the same genes exhibited normal cellular growth (Fig. 3d). When we quantified the mRNA level of these genes, it was observed that the Tl-CRISPRi induced only a moderate or weak decrease of mRNA level, which ranges from 40% to 70% compared to the non-target control strain (Supplementary Fig. 7a). We speculated that this moderate decrease of mRNA contributed to a less significant polar effect in contrast to the Tx-CRISPRi. Additionally, to verify that Tl-CRISPRi efficiently knocked down targeted endogenous genes in the translation level, we fused *nanoluciferase* with a flexible linker to the C-terminal region of the target gene to measure the expression level. When the Tl-CRISPRi was adopted, the luminescence level remarkably decreased compared to the non-targeted control (Supplementary Fig. 7b), confirming that Tl-CRISPRi robustly and specifically knocked down the expression of these genes while circumventing the polar effect. Overall, our Tl-CRISPRi system could be exploited for the specific down-regulation of each cistron in the synthetic and native bacterial operons, exerting diminished interference with the expression of neighboring genes in the same transcription unit.

## Engineering the handle sequence of guide RNA for tunable repression by Tl-CRISPRi

Several gene expression modulators, such as dCas9 or synthetic sRNA-Hfq, have been engineered for fine-tuning gene expression levels rather than ON/OFF control, especially for metabolic engineering applications[36–40]. We considered establishing an appropriate fine-tuning strategy for Tl-CRISPRi could be beneficial to expand its usage. Although modulating the expression level of dCas13 could be one way of achieving dose-dependent gene knockdown[41] (Supplementary Fig. 2), the gene regulation dynamics could be inherently affected by the promoter strength of *dCas13* or cell-to-cell variation of the inducible gene expression system[38,42]. Introducing mismatch to the spacer

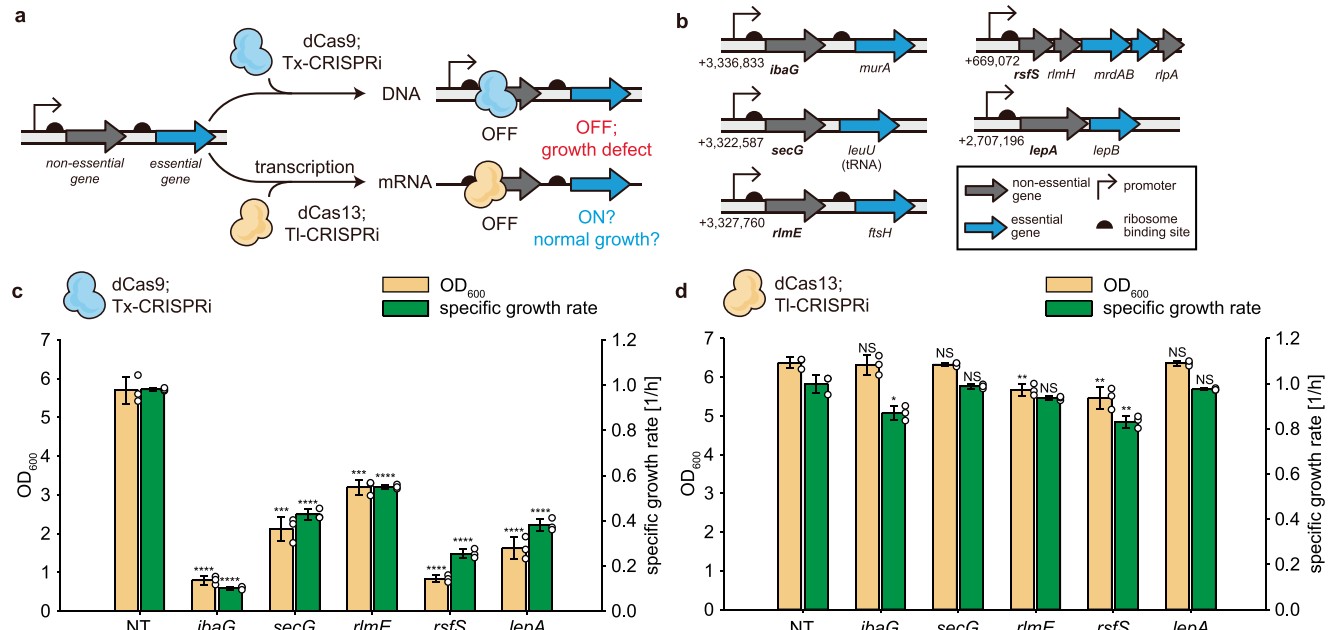

**Fig. 3 | Diminished polar effect and growth retardation by the Tl-CRISPRi while targeting the endogenous operons of *E. coli*. a** A diagram shows the difference between the Tx-CRISPRi and the Tl-CRISPRi while targeting the operon of non-essential genes and essential genes in series. Though the non-essential gene was targeted, the polar effect causes Tx-CRISPRi to block the expression of downstream essential genes, thus hindering bacterial growth. We speculated that the Tl-CRISPRi would not cause growth retardation while targeting the same non-essential genes. **b** Five different endogenous operons on the *E. coli* K-12 MG1655 chromosome were selected as examples, and their transcription unit is presented. **c**, **d** The cellular growth while performing Tx-CRISPRi (**c**) or Tl-CRISPRi (**d**) toward the upstream non-essential genes was compared with the strain harboring the non-target guide RNA (NT). The error bar represents the mean ± standard deviation from the biologically independent cell cultures (*n* = 3), and the white dots indicate the actual data points. The *P*-value of each strain's dataset was determined by the two-tailed Student's *t*-test compared to the dataset of the NT strain. The asterisk above the bar indicates the *P*-value. NS: not significant; **P* < 0.05, ***P* < 0.01, ****P* < 0.001, *****P* < 0.0001. Source data are provided as a source data file.

region was another strategy for fine-tuning the repression strength in the Tx-CRISPRi[43,44]. However, estimating the activity of mismatched guide RNA is complicated due to the differed tolerance of mismatch position across the spacer, requiring massive experimental datasets and modeling[45]. In our previous study, a library of sgRNA of dCas9 generated by mutating tetraloop and flanking regions yielded sgRNA variants with diversified repression efficiencies[39]. This study suggested that modulating the efficiency of RNP complex formation by mutating the handle sequence of the guide RNA could control the repression efficiency of CRISPRi. In line with this approach, we explored whether mutating the direct repeat (DR) sequences of the guide RNA, known to be recognized by the Cas13 protein[29,46], could achieve tunable expression of a target gene in the Tl-CRISPRi system (Fig. 4a). From the structure study, it was revealed that the stem distal from the spacer and the terminal loop is solvent-exposed and does not interact with the residues of Cas13d, while the stem proximal to the spacer and nucleotides comprising the flanking sequences directly interact with the Cas13d[29,46]. Based on these findings, we selected potential sites of mutations in the guide RNA handle by dividing it into three parts: stem, bulge, and flanking sequence (Fig. 4b). These parts were systematically mutated, creating guide RNAs with diverse DR sequences. We applied them to knock down *mCherry* and measured the relative expression level derived from each mutated guide RNA (Fig. 4a).

For the mutations in the stem of DR, disrupting base pairings yielded plenty of attenuated guide RNAs inducing diverse expression levels of mCherry (Supplementary Fig. 8a). However, replacing GC base pairing into AU did not perturb the repression strength, albeit increasing the minimum free energy of the stem-loop (Supplementary Fig. 8b). When the length of the stem was varied from 4 bp to 9 bp, shortening and extending the stem for more than 2 bp from the original DR significantly weakened the repression of *mCherry*

(Supplementary Fig. 8c). Considering that extending the stem duplex helps to stabilize the RNA and increases the knockdown efficiency in the case of antisense RNA[47], it was surprising that the stem longer than 8 bp strongly attenuated the knockdown of *mCherry* compared to the original DR. This suggests that the folding energy and the stability of guide RNA itself is not the sole factor that affects the repression strength of Tl-CRISPRi. Indeed, the minimum free energy of each stem-mutated DR and the relative expression level of mCherry derived from DR-mutated guide RNAs did not show a remarkable correlation (Supplementary Fig. 9). Instead, we reasoned that the interaction between DR of guide RNA and the residues of dCas13 would play an important role determining the knockdown strength of Tl-CRISPRi, which should be explored in further studies. Unlike the mutations on the stem, modifying the bulge did not generate significantly attenuated guide RNAs, implying that these bases would not or weakly interact with the residues of dCas13 (Supplementary Fig. 10). Substituting the base of flanking sequences adjacent to the stem (U25, G26) also generated attenuated guide RNAs (Supplementary Fig. 11), suggesting that they could be the potential targets for guide RNA attenuation. We obtained 51 different handle-mutated guide RNAs that can derive target gene expression levels in a uniform distribution from 2.6% to 86.3% compared to the non-target control (Fig. 4c).

To investigate whether the mutated DR derives a similar relative expression level across other genes, we targeted *GFP* and *nanoluciferase* with the DR-mutated guide RNAs. The relative expression derived from the same mutated DR showed strong positive correlations across different reporter genes (Fig. 4d). These results validate that our tunable Tl-CRISPRi system could be adapted to fine-tune the expression level of various genes in *E. coli*. We then applied our tunable Tl-CRISPRi system in *Vibrio natriegens*, a fast-growing and metabolically versatile bacterial strain[48]. When the guide RNA with the original

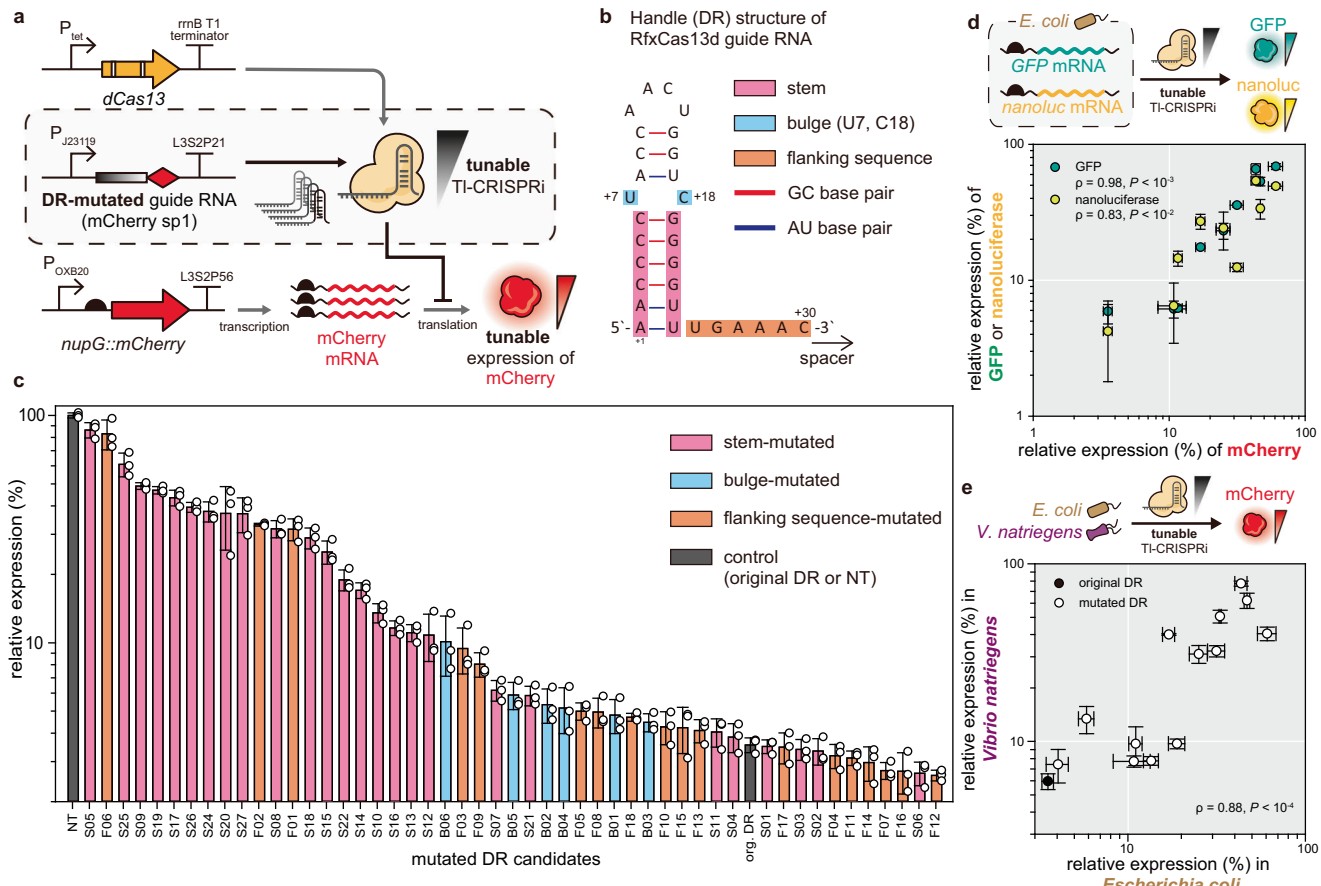

**Fig. 4 | Development and application of tunable Tl-CRISPRi based on the handle-engineered guide RNAs. a** Genetic components for the validation of tunable Tl-CRISPRi system. The DR sequence was mutated to modulate the interaction between the dCas13 and the guide RNA. This tunable Tl-CRISPRi was directed to inhibit the translation of mCherry mRNA in diverse levels, leading to the tunable expression of mCherry. **b** The secondary structure of the guide RNA handle. This structure was divided into bulge, stem, and flanking sequence. Each part's bases were mutated, creating candidate DR sequences for the attenuated guide RNA. **c** The resulting expression levels of mCherry derived from the guide RNAs harboring diverse DR sequences are shown in the graph. **d** Relative GFP and nanoluciferase expression level against the mCherry while using the guide RNA with the same mutated DR sequence. GFP sp2 and nanoluc sp1 were each adopted for the knockdown of *GFP* and *nanoluciferase* as a spacer sequence. All of the reporter genes were expressed from the chromosome of *E. coli*. **e** The relative expression level of mCherry in the *V. natriegens* compared to that of *E. coli* when the same attenuated guide RNAs were adopted. The Spearman's correlation coefficient (ρ) and the related *P*-value determined by the two-tailed Student's *t*-test are annotated in each graph (**d**, **e**). The relative expression was determined based on the specific fluorescence level (RFU/OD$_{600}$) and normalized by setting the value of the NT (non-targeted) strain as 100%. The error bar represents the mean ± standard deviation from the biologically independent cell cultures ($n = 3$), and the white dots indicate the actual data points. Source data are provided as a source data file.

DR was used to target mCherry mRNA, the expression level of mCherry decreased to 6.0%, showing that the Tl-CRISPRi could effectively repress genes in *V. natriegens* (Fig. 4e). Furthermore, the guide RNAs containing mutated DR sequences regulated the expression of mCherry at various levels, with a positive correlation to that of *E. coli* (Fig. 4e). Our tunable Tl-CRISPRi system based on the engineered guide RNAs has the potential to precisely down-regulate the gene expression in other microbial systems beyond *E. coli* and *V. natriegens*.

**Applying Tl-CRISPRi to optimize the metabolic flux**
Optimizing the metabolic flux toward producing valuable compounds by specific knockdown or knockout of genes is one of the common approaches in metabolic engineering[8,49]. We examined whether the Tl-CRISPRi could enhance metabolite production by adjusting the endogenous gene expression level and, thus, intracellular metabolic flux. We selected the pathway converting malonyl-CoA into 3-HP[50], a valuable industrial platform chemical[51]. This pathway is advantageous in that the malonyl-CoA is a universal metabolic intermediate, which allows the utilization of diverse carbon sources[52,53]. We employed *E. coli* W (ATCC 9637) for the bacterial host, possessing several beneficial

characteristics for metabolite production, such as acid tolerance and reduced byproduct[54]. We selected 48 genes and targeted them using the guide RNAs with the original DR sequence to ensure effective gene repression and determine which gene target knockdown could enhance the titer of 3-HP (Fig. 5a).

Since this pathway consumes 2 moles of NADPH per 1 mol of malonyl-CoA to produce 3-HP, we reasoned that targeting genes related to the NADPH or malonyl-CoA pool could affect the bioproduction of 3-HP. Indeed, repressing genes comprising the fatty acid synthesis cycle, a competitive pathway for consuming malonyl-CoA, significantly increased the 3-HP production (Fig. 5b). The knockdown of the *fabI* gene showed the most prominent increase of 3-HP titer up to 1.87 g/L, which was 11.1-fold higher than the non-targeted strain (0.17 g/L). This result could be attributed to the characteristics of the enoyl-acyl carrier protein reductase encoded by *fabI*, which holds the most crucial step for completing the fatty acid synthesis cycle[55]. In glycolysis, the knockdown of *aceE*, *pfkA*, and *pfkB* enhanced the 3-HP titer (Fig. 5b). At first, it seemed controversial that the knockdown of *aceE* encoding one of the subunits of pyruvate dehydrogenase complex (PDC) was beneficial for the 3-HP production since the

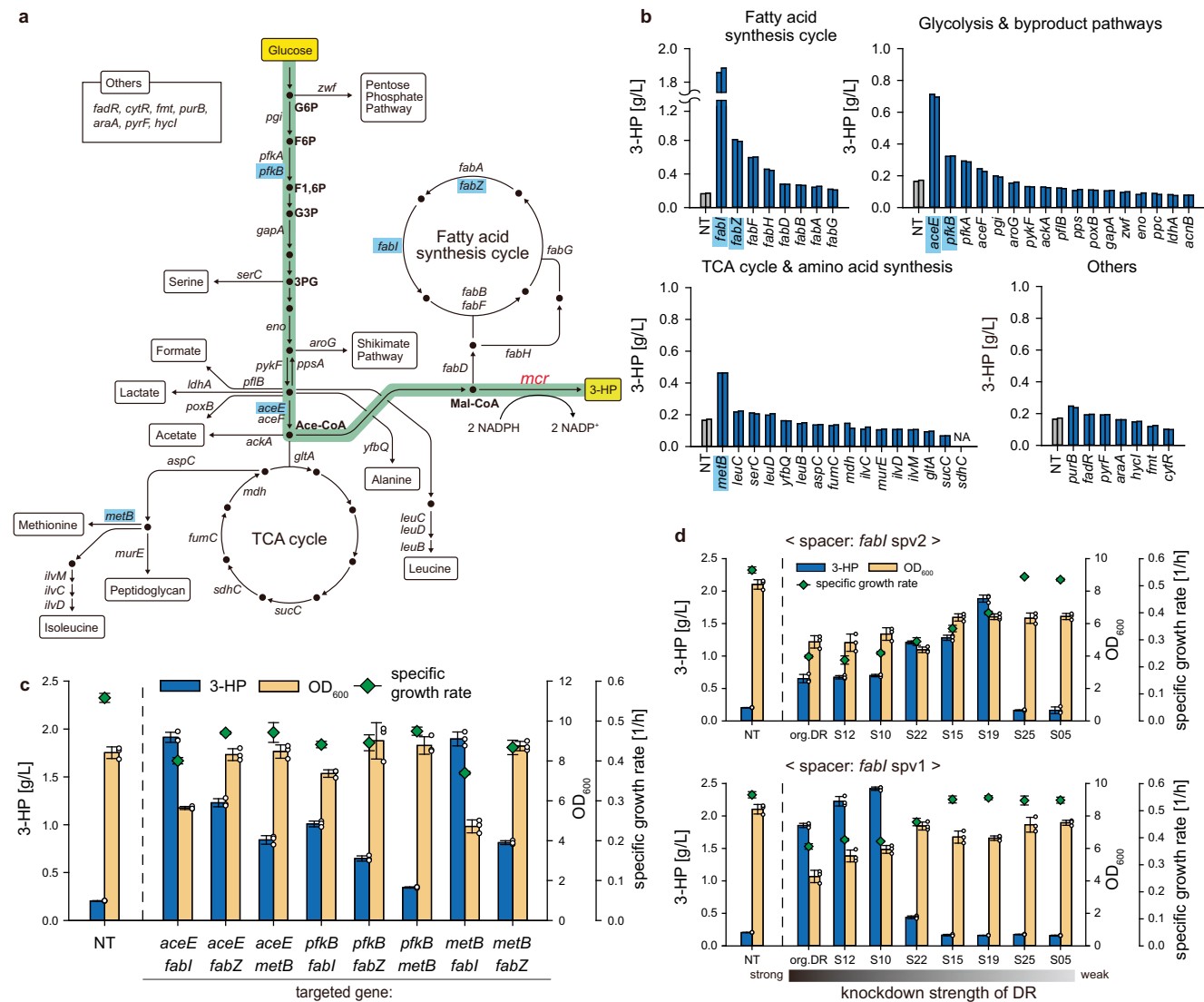

**Fig. 5 | Metabolic pathway redirection via tunable Tl-CRISPRi for enhancing the production of 3-HP.** All of the experiments shown in this figure were performed based on the acid-resistant strain *E. coli* W. **a** Overall pathway for synthesizing 3-HP from the glucose by utilizing malonyl-CoA reductase (*mcr*) and its neighboring pathways (TCA cycle, fatty acid synthesis cycle, etc.). All of the target genes of the Tl-CRISPRi were annotated in this figure. **b** The 3-HP titer of each strain when the Tl-CRISPRi knocked down each gene target. We selected five genes (*fabI, fabZ, aceE, pfkB, metB*), of which the knockdown exhibits a notable improvement of 3-HP titer,

for further experiments. **c** The Tl-CRISPRi simultaneously knocked down two genes from the five selected candidates in (**b**). **d** To fine-tune the expression level of the *fabI* gene, 16 different guide RNAs consisting of eight different DR sequences and the two *fabI* spacers (spv1 and spv2) were adopted. The 3-HP titer and the OD$_{600}$ were measured after 24 h of the induction of the *mcr* gene. The error bar represents the mean ± standard deviation from the biologically independent cell cultures (*n* = 3), and the white dots indicate the actual data points. Source data are provided as a source data file.

knockdown or knockout of PDC reduces the flux toward the acetyl-CoA[56,57]. However, in one previous study, it was revealed that the knockout of the *lpdA* gene encoding another component of PDC could shift the carbon flux of glycolysis from the Embden−Meyerhof−Parnas (EMP) pathway to the Entner−Doudoroff (ED) pathway or Pentose Phosphate (PP) pathway, thereby increasing the intracellular level of NADPH[58]. The knockdown of the *pfkA* or *pfkB* gene would also enhance the 3-HP titer by redirecting the carbon flux toward ED and PP pathway[59,60]. Among the knockdown targets in amino acid biosynthesis pathways, the knockdown of *metB* related to the generation of methionine resulted in 3-HP titer enhancement (Fig. 5b). We reasoned that the knockdown of *metB* would increase the intracellular NADPH pool since the synthesis of methionine consumes the most NADPH molecule per one amino acid[61,62]. In a comprehensive approach, we employed combinatorial gene knockdown to investigate the collective effect of these knockdown targets on enhancing 3-HP production

(Fig. 5c). However, the simultaneous knockdown of *fabI* with other genes did not increase the 3-HP titer against the single knockdown of *fabI*. Repressing the *aceE* and *fabZ* showed combinatorial enhancement of 3-HP production from 0.70 g/L and 0.80 g/L under the single knockdown to 1.23 g/L under the double knockdown without causing growth retardation.

Notably, the knockdown of *fabI* significantly decreased microbial growth (Fig. 5c, d). Although blocking the synthesis of fatty acid and preserving the pool of malonyl-CoA was beneficial for 3-HP production, fully repressing this pathway could hamper microbial growth and limit the optimal fermentation process[63]. Therefore, we tried to adopt the attenuated guide RNAs for the moderate repression of the *fabI* gene and the optimal bioproduction of 3-HP. To implement diverse knockdown strength on the *fabI*, we designed another spacer (*fabI* spv2) targeting more upstream of the *fabI* mRNA than the original spacer (*fabI* spv1) and adopted mutant DR sequences to both spacers,

yielding 16 different guide RNAs to repress *fabI* in total. We found that the cellular growth gradually increased as the weaker DR sequence was utilized in both spacers (Fig. 5d). It was also noticeable that the optimal DR sequence reaching a maximal 3-HP production differed in these two spacers (S19 in *fabI* spv2 and S10 in *fabI* spv1). In total, the titer of 3-HP reached the maximum of 2.42 g/L when the guide RNA with S10 DR and *fabI* spv1 spacer was used, which was 14.2-fold improved against the strain with non-target guide RNA (Fig. 5d). However, the titer of 3-HP was suddenly dropped as the weaker DR sequence was adopted for both spacers. These results demonstrate that adjusting the expression level of *fabI* in a fine-tuned manner for securing an adequate amount of malonyl-CoA was critical for diminishing growth retardation and achieving the optimal production of 3-HP. Since the malonyl-CoA serves as a versatile precursor for the biosynthesis of diverse value-added compounds such as polyphenols and polyketides[64], we anticipate that our tunable Tl-CRISPRi system capable of modulating the fatty acid synthesis could be further expanded toward the production of these compounds.

Since all of these Tl-CRISPRi experiments on optimizing 3-HP production were conducted based on the small-scale test tube culture with a small amount of glucose (8 g/L), we investigated whether the effect of the Tl-CRISPRi repression could be maintained for a larger scale of culture and a higher concentration of carbon source. We performed flask-scale fed-batch culture using the best-regulated strains from the Tl-CRISPRi screening, starting from 20 g/L of glucose. We observed that the 3-HP titers were also remarkably increased in these strains compared to the non-targeted strain, which was 16.4-fold higher in strain with S10-*fabI* spv1 guide RNA (Supplementary Fig. 12). This result suggests that our tunable Tl-CRISPRi could robustly function and redirect the native metabolic flux of microbial cell factories.

## Discussion

Repurposing the CRISPR-Cas system as a robust synthetic biology tool has boosted genome engineering, metabolic flux redirection, and the discovery of genotype-phenotype relationships in numerous microbial systems[33,34,65]. The innate programmability of the CRISPR-Cas system allows flexible targeting of genes and expansion of functions by fusing diverse protein effectors[6,66,67]. The advent of dCas13 from the Type VI CRISPR-Cas system holds new promise as a synthetic bacterial gene regulation tool, as it specifically recognizes and binds to the RNA of interest rather than the DNA-targeting dCas9 or dCas12[9,10,19].

In this study, we showed that detailed characterization of the Tl-CRISPRi system with dose-dependent, specific, multiplexed gene knockdown in translation level would accelerate the construction of a dCas13-based genetic circuit. We observed that only the reporter genes targeted by the guide RNAs were specifically knocked down in multiplexed gene targeting, which suggests the high specificity of our system. However, due to the difficulties in performing proteome analysis, we could not inspect the genome-wide off-target perturbations of translational inhibition under the application of Tl-CRISPRi. Further investigation on proteome-level perturbation caused by Tl-CRISPRi could suggest a detailed understanding of the off-target effect and propose design rules of guide RNA to enhance specificity.

While targeting polycistronic operons, we also showed that Tl-CRISPRi could overcome the polar effect and independently repress the expression of each cistron in an operon, validated in the reporter gene operons and native bacterial operons. The polar effect caused by the Tx-CRISPRi is one of the obstacles to the accurate identification of the role of each gene in a polycistronic operon while performing pooled genome-wide Tx-CRISPRi screening[34]. We anticipate that additional genome-wide Tl-CRISPRi screening could reveal unprecedented information about the genotype-phenotype relationships in bacterial cells not discovered in the Tx-CRISPRi-based screening.

Based on the well-characterized Tl-CRISPRi system, we developed a fine-tuning strategy by engineering the handle structure of the guide RNA and modulating the interaction between the guide RNA and dCas13. Though we focused on the down-regulation of gene expression level, our attenuated guide RNAs might be applied to achieve diverse magnitudes of gene activation if the dCas13 is fused with translational activators[13]. With the tunable Tl-CRISPRi system, we achieved a 14.2-fold improvement of 3-HP production by precisely redirecting metabolic flux without laborious genome engineering and replacing genetic parts. Since our fine-tuning strategy does not require adjusting the expression level of the dCas13 or the guide RNA, we expect that our tunable Tl-CRISPRi would help tune gene expression and optimize the metabolic network in biological systems where the usage of inducible gene expression systems or precisely characterized genetic parts are limited[68]. Also, targeting multiple pathways with different down-regulation strengths by assembling multiple guide RNAs with diverse mutant DR sequences could be applied to diversifying the expression of genetic circuits and metabolic flux. Analyzing the resulting bioproduction via machine learning could suggest further design rules to guide RNAs and accelerate automated genetic engineering to approach the optimum of the genetic circuit and metabolic flux[69,70].

While we applied the tunable Tl-CRISPRi in two gram-negative bacterial strains, detailed characterization on bacterial strains other than *E. coli* should proceed to validate the expandability of our system toward diverse strains. We believe that further investigation and optimization could enable the usage of this tool in diverse bacterial systems, including gram-positive bacteria, as other CRISPR-based tools have been successfully adopted and widely utilized in diverse strains[71,72].

To summarize, we developed a tunable Tl-CRISPRi system based on the dCas13 and systematically engineered guide RNAs. We demonstrated successful knockdown of target genes transcribed from the chromosome and the plasmid, together with the sequence-specificity and multiplexity of gene repression. We also highlighted the uniqueness of Tl-CRISPRi versus Tx-CRISPRi in that it allows context-independent regulation of each cistron in an operon. To expand the scope of gene regulation, we established a fine-tuning strategy for the Tl-CRISPRi by engineering the handle sequence of the guide RNA. This tunable Tl-CRISPRi exhibited its usefulness for redirecting the metabolic flux in a fine-tuned manner and optimizing the bioproduction of a valuable compound, 3-HP. Further expansion of the tunable Tl-CRISPRi to optimize the production of diverse metabolites in non-model bacterial strains would expedite the engineering and development of undomesticated microorganisms with versatile and unique metabolic traits[2,72].

## Methods

### Bacterial strains, plasmids, primers, and reagents

The plasmids and the bacterial strains used in this study are listed in Supplementary Data 1. Oligonucleotides used in this study were synthesized by Bionics, and the names and the sequences are listed in Supplementary Data 2. Luria–Bertani (LB) broth and Bacto-agar for bacterial culture were purchased from BD Bioscience. Plasmid DNA from cloning cells was prepared using the Exprep™ Plasmid SV mini kit (GeneAll). Polymerase chain reaction (PCR) fragments were purified using DNA Clean & Concentrator Kit (Zymo Research) or the Expin™ Gel or PCR SV Kit (GeneAll). For the gene cloning, Q5 polymerase, Quick ligase, T4 PNK, and restriction enzymes were purchased from New England Biolabs, and Pfu polymerase was purchased from Bioneer. Gibson assembly was performed using Gibson Assembly® HiFi master mix from CODEX DNA.

The bacterial strains harboring reporter gene expression cassette (*GFP*, *mCherry*, *nanoluciferase*) were prepared using the λ-Red recombination system[73]. After overnight culture and refreshing, *E.*

coli K-12 MG1655 harboring pSIM5[73] plasmid was grown for 2 h at 30 °C with shaking (250 revolutions per min (rpm)). It was shifted to 42 °C for 15 min to induce the expression of the λ-Red recombinases. After that, the cells were centrifuged and washed to make electro-competent cells. Four hundred microgram of DNA editing template containing FRT-kanR-FRT and the reporter gene expression cassette was electroporated into these cells for recombination. The kan^R gene should be removed from the chromosome for subsequent recombination of the other reporter genes. To this end, the pCP20 plasmid was transformed into recombinant E. coli, and the flippase was induced at 37 °C during overnight culture to remove FRT-kan^R from the chromosome.

## Construction of genetic components for the Tl-CRISPRi system

Mach-T1^R cells were used for plasmid construction and cultured in LB medium containing appropriate antibiotics. To construct the plasmids with different dCas13 orthologs, the backbone plasmid containing aTc inducible gene expression cassette was prepared by assembling two PCR fragments. One fragment containing tetR, bidirectional $P_{tet}$ promoter, and the ribosome binding site of the dCas13 gene was amplified by tetR_BciVI_F and Ptet_RBS_GA_R. The other fragment containing the rrnB T1 terminator was amplified by rrnBT1_RBS_F and rrnBT1_pACYC_GA_R. These fragments were assembled into the pACYCDuet backbone digested by BciVI and Bsu36I, yielding the pACYC_tetR plasmid. Several dCas13 genes were incorporated between the ribosome binding site and the rrnB T1 terminator of the pACYC_tetR. For generating a catalytically dead version of BzCas13b and PbCas13b, three separate PCR products, each containing alanine substitution mutations in the catalytic residues of the HEPN domain, were assembled and incorporated into the pACYC backbone using Gibson assembly. The original Cas13 or dCas13 genes are amplified from p2CT-His-MBP-Lbu_C2c2_R472A_H477A_R1048A_H1053A (Addgene #83485), pBzcas13b (Addgene #89898), pPbCas13b (Addgene #89906), and pXR002: EF1a-dCasRx-2A-EGFP (Addgene #109050).

The precursor CRISPR-RNA (pre-crRNA) expression plasmid was constructed based on pCDF-sgRNA-BsaI. To construct the LbuCas13a crRNA expression plasmid, two different PCR fragments were prepared; one is amplified by using SmR_BclI_F and crRNA_LbuCas13a_DR1_R, and the other is amplified by using SmR_BclI_R and crRNA_LbuCas13a_DR2_F. These two fragments were cut by BclI and ligated, yielding pCDF_precrRNA(Lbu)-NT. The other pre-crRNA expression cassettes were constructed similarly. For creating the vector containing a matured version of RfxCas13d guide RNA, pCDF_sgRNA-BsaI was PCR amplified with RfxCas13d_matureRNA_BsaI_F and RfxCas13d_matureRNA_BsaI_R and self-ligated. To add a strong terminator downstream of mature guide RNA, the L3S2P21 terminator[74] was amplified by L3S2P21_Bsu36I_F and L3S2P21_AgeI_R and inserted between the Bsu36I and AgeI sites of pCDF_RfxgRNA-NT. Fragments harboring spacer sequences with appropriate 4-bp overhang were prepared by PNK treatment followed by oligo annealing, and they were cloned into the pCDF_RfxgRNA-NT vector cut by BsaI. For the effective targeting of dCas13, the spacer sequences were rationally selected to form the least secondary structure inside itself while targeting the translation initiation region spanning from the Shine-Dalgarno sequence to the start codon of a target gene. The spacer sequences used in this study are listed in Supplementary Data 3.

The pACYC plasmid harboring dRfxCas13d and constitutively expressed mCherry was prepared from the pACYC_dRfx. The mCherry gene expression cassettes with four different constitutive promoters were amplified by TJmT_pACYC_GA_F and mCherryRBS_J231XX_GA_R. The PCR fragment harboring the L3S2P21 terminator was amplified by TJmT_Pr_GA_F and L3S2P21_TJmT_GA_R. These two PCR fragments were assembled with the backbone plasmid using Gibson assembly and cloned into the downstream of tetR gene. The plasmids carrying dRfxCas13d, mCherry, and the guide RNA, which was utilized for the validation of tunable Tl-CRISPRi in V. natriegens, were prepared from the pACYC_dRfx_$P_{J23100}$_mCherry. The guide RNA expression cassette was amplified from pCDF-RfxgRNA vectors harboring the guide RNA of interest by L3S2P21_pACYC_GA_R and UPJ23119_TJmT_GA_F and was assembled into the pACYC_dRfx_$P_{J23100}$_mCherry digested by BamHI.

The mutated DR sequences of attenuated guide RNAs were altered by site-directed mutagenesis. For example, the pCDF_RfxgRNA-mCherry sp1 was PCR amplified using PJ23119_DRvariant (S01–F18)_R containing the mutant DR sequence, and mChsp1_invPCR_F, followed by PNK treatment and self-ligation by Quick Ligase. Other attenuated guide RNAs with different spacer sequences were prepared using the corresponding primers for each spacer and the mutated DR. The mutated DR sequences used in this study are listed in Supplementary Data 3.

The pET_Ptac-mcr for the IPTG-inducible expression of the mcr gene in E. coli W was prepared by the following steps. The functionally enhanced mutant mcr gene[75] encoding MCR^N940V/K1106W/S1114R was amplified from pET-mcr* plasmid[52] with the Ptac_EcoNI_F and mcr_AvrII_R. This PCR fragment was digested by EcoNI and AvrII and ligated into pETDuet plasmid cut by the same restriction enzymes.

## Measuring relative expression levels of reporter genes and the specific growth rates in E. coli

All culture experiments measuring fluorescence or luminescence in E. coli were performed using an LB medium containing appropriate antibiotics (34 μg/mL for chloramphenicol and 50 μg/mL for spectinomycin). The overnight inoculum from a single colony was diluted 1/100 into the fresh media with 100 ng/mL of aTc except in the experiment of Supplementary Fig. 2, which was performed by varying the concentration of aTc. After incubating for 8 h at 37 °C with 250 rpm shaking, 100 μL of culture was sampled and washed with the PBS buffer, and the fluorescence was detected using a Hidex Sense microplate reader (Hidex). The luminescence from nanoluciferase was generated using the Nano-Glo® Luciferase Assay System from Promega and was detected by Hidex. For dilution, the nanoluciferase buffer, PBS, cell culture, and the nanoluciferase substrate were mixed with a ratio of 50:40:10:1. $OD_{600}$ was measured from culture broth using the Jenway 7300 spectrophotometer. The relative expression was determined by dividing fluorescence or luminescence units into $OD_{600}$ values and setting the value for the non-target control strain as 100%. For measuring the maximum specific growth rate of E. coli under different conditions, the overnight culture of each colony was refreshed into a ratio of 1:100 with 100 ng/mL of aTc and loaded on the 96-well microplate, which was incubated at 37 °C with 600 rpm shaking using a Hidex. The maximum specific growth rate was quantified from linear regression of logarithmic $OD_{600}$ during the exponential phase. The $OD_{600}$ data of time points between 1 h and 3 h after refreshing the culture were used to calculate specific growth rates, except for the experiment of Fig. 5, which used the data of 2–4 h after refreshing.

## Analyzing protein expression by western blotting and coomassie blue staining

To analyze the expression of four different dCas13 orthologs, the 6x His-tag was inserted at the N-terminal of each dCas13 protein by site-directed mutagenesis using Lb, Bz, Pb, Rfx_NHis(6x)_F and dCas13_N-His(6x)_R. The NHis-dCas13 plasmid and the cognate guide RNA plasmid expressing pre-crRNA targeting mCherry mRNA were transformed into EcM-mCherry cells. Culture conditions were the same for measuring the fluorescence of mCherry. After 8 h of induction, cells reached the density of 1.8 $OD_{600}$ were collected by centrifugation and resuspended into a 300 μL volume of either lysis buffer (100 mM $NaH_2PO_4$, 10 mM Tris-HCl, and 8 M urea, pH 8.0) for the total fraction or BugBuster® Protein Extraction Reagent (Merck Millipore) for the soluble fraction. The lysates were sonicated by Q800R Sonicator (Qsonica) and were loaded in SurePAGE™ 4–12% Bis-Tris gels (GenScript) with MOPS SDS Running Buffer (Thermo). In order to visualize

whole protein, the gel was incubated with SimplyBlue™ SafeStain (Invitrogen) for 1 h and destained with deionized water overnight. For the visualization of His-tagged dCas13, western blotting was performed using the invitrolon PVDF/filter paper sandwiches (Invitrogen), 6×-His tag monoclonal antibody (eBioscience), anti-mouse IgG (whole molecule)-alkaline phosphatase antibody produced in goat (Sigma) and 1-step™ NBT/BCIP substrate solution (Thermo) following the instruction of manufacturers. The 6×-His tag monoclonal antibody was diluted to 3:10,000, and the anti-mouse IgG (whole molecule)-alkaline phosphatase antibody was diluted to 1:10,000 during the antibody incubation steps.

### Measuring relative expression levels of mCherry in *V. natriegens*

All culture experiments measuring fluorescence in *V. natriegens* were performed using an LBv2 medium (LB medium supplemented with 11.9 g/L of NaCl, 2.2 g/L of $MgCl_2$, and 0.31 g/L of KCl) containing appropriate antibiotics (10 μg/mL for chloramphenicol). The overnight inoculum from a single colony was diluted to the $OD_{600}$ of 0.05 into the fresh media with 100 ng/mL of aTc. After incubating for 18 h at 30 °C with 250 rpm shaking, 100 μL of culture was sampled and washed with the PBS buffer, and the fluorescence was detected using a Hidex Sense microplate reader (Hidex). $OD_{600}$ was measured from culture broth using the Jenway 7300 spectrophotometer. The relative expression was determined by dividing relative fluorescence units into $OD_{600}$ and setting the value for the non-target control strain as 100%.

### RNA extraction and one-step reverse transcription and quantitative PCR

The relative amount of mRNA of mCherry, GFP, and nanoluciferase was determined by real-time quantitative PCR (RT-qPCR). The total RNA of each strain was prepared from the cell culture by the same condition applied for fluorescence and luminescence measurements. The RNeasy Plus Mini Kit (Qiagen) was used for RNA extraction, and the on-column DNase treatment was followed by the RNase-free DNase set (Qiagen). The concentration of the total RNA sample was measured by NanoDrop One (Thermo Scientific), and the extraction quality was verified using DNA 5K/RNA/CZE24 LabChip (PerkinElmer) on a LabChip GX Touch 24 (PerkinElmer). The one-step reverse transcription and quantitative PCR were performed using Luna® Universal one-step RT-qPCR kit (New England Biolabs) and StepOnePlus real-time PCR system (Applied Biosystems). 180 ng of total RNA sample was used for 10 μL of qPCR mixture. The oligonucleotide primers used for the RT-qPCR experiment are listed in Supplementary Data 2. Three reference genes, *cysG*, *idnT*, and *hcaT*, were combinatorially used as internal standards of RT-qPCR experiments[76]. The relative target mRNA quantity was calculated by the ddCt method[77] with the help of StepOnePlus Software (Applied Biosystems).

### Measuring knockdown efficiencies of endogenous operons of *E. coli* by the Tl-CRISPRi

The strains harboring the *nanoluciferase*-fused endogenous gene were prepared using pCas_CDF for scar-less recombination[78]. After overnight culture and refreshing, the *E. coli* K-12 MG1655 strain harboring pCas_CDF was grown for 2 h at 30 °C with 0.2% arabinose to induce the expression of λ-Red recombinases. The cells were washed with distilled deionized water for making electro-competent cells and electroporated with 200 ng of the appropriate pCDF_sgRNA and 600 ng of dsDNA editing template. After checking successful recombination and detecting the activity of luciferase, subsequent curing of the pCDF plasmid was performed by culturing in LB with 50 μg/mL of kanamycin and 200 μM of IPTG at 30 °C. pCas_CDF was cured by culturing in LB at 37 °C. Subsequently, pACYC_dRfx and pCDF plasmid containing appropriate guide RNA were electroporated for the Tl-CRISPRi

experiment. The time point of sampling was 6 h after refreshing the culture and induction of the dCas13. The luminescence was generated using the Nano-Glo® Luciferase Assay System from Promega and was detected by Hidex. The nanoluciferase buffer, PBS, cell culture, and the nanoluciferase substrate were mixed with a ratio of 50:40:10:1.

### Analysis of 3-HP production in *E. coli* W

We transformed pACYC_dRfx, pCDF_RfxgRNAs with the effective spacer, and pET_Ptac-*mcr* in the *E. coli* W (ATCC 9637) cell. The cell culture and 3-HP production was performed in modified M9 media (8 g/L of glucose, 2 g/L of NaCl, 2 g/L of $NH_4Cl$, 10.7 g/L of $K_2HPO_4$, 5.2 g/L of $KH_2PO_4$, 0.5 g/L of $MgSO_4 \cdot 7H_2O$, and 1 g/L of yeast extract) containing appropriate antibiotics (34 μg/mL for chloramphenicol, 50 μg/mL for spectinomycin, and 75 μg/mL for ampicillin), at 37 °C with 250 rpm shaking. For the experiments in Fig. 5, we conducted the bacterial culture in the test tube with a 2 mL volume of media. The overnight culture was diluted at an $OD_{600}$ of 0.05 and induced with 100 ng/mL of aTc for the expression of dCas13. Subsequently, when the $OD_{600}$ of the culture was reached at 0.8, 20 μM of IPTG was added to induce the *mcr* gene. After 24 h of shaking incubation, the culture broth was centrifuged, and the supernatant was sampled to measure the concentration of 3-HP. The fed-batch fermentations in Supplementary Fig. 12 were performed in the 25 mL of culture volume in the 250 mL baffled flask and a 20 g/L initial concentration of glucose. The overall scheme of fermentation was the same with the test tube culture except for the concentration of IPTG, which was 200 μM in the fed-batch fermentation. The HPLC analysis for the samples was performed by a 1260 Infinity II LC system (Agilent) equipped with a Hi-Plex-H column (Agilent). The flow rate, mobile phase, and column temperature were 0.6 mL/min, 5 mM of $H_2SO_4$, and 40 °C, respectively.

### Reporting summary

Further information on research design is available in the Nature Portfolio Reporting Summary linked to this article.

## Data availability

All data supporting the findings of this work are available within the paper and its Supplementary Information file. Bacterial strains and plasmids used in this study, and the other datasets generated and analyzed during this study are available from the corresponding author upon request. Source data are provided with this paper.

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

## Acknowledgements

This work was supported by the Bio & Medical Technology Development Program (NRF-2021M3A9I4024737, NRF-2021M3A9I5023245, and RS-2024-00352569) and a grant (RS-2024-00345885) of the National Research Foundation (NRF) funded by the Korean government (MSIT). We also acknowledge that this work was supported by the Korea Institute of Marine Science & Technology Promotion (KIMST) funded by the Ministry of Oceans and Fisheries [20220258]. Hyeon Jin Kim was supported by the Hyundai Motor Chung Mong-Koo fellowship.

## Author contributions

G.K. and S.W.S. designed this project and overall experiments. G.K. and H.J.K[1]. constructed genetic systems and microbial strains. G.K. performed the fluorescence and luminescence measurements, RT-qPCR, and HPLC analysis of *E. coli*. K.K. cultured *V. natriegens* strains and analyzed the fluorescence of them. G.K., Ho Joon Kim, K.K., Hyeon Jin Kim, J.Y., and S.W.S. participated in the data analysis and discussions. G.K. and S.W.S. wrote the manuscript, and all authors read and approved the final manuscript.

## Competing interests

G.K. and S.W.S. are inventors on Korean patent application 10-2023-0151135 and PCT patent application PCT/KR2024/006932, filed by Seoul National University Research and Development Foundation. This patent covers the tunable Tl-CRISPRi technology via mutating the DR of guide RNA. All other authors declare no competing interests.
