## [Peer Review File · Nature Communications]

Tunable translation-level CRISPR interference by dCas13 and engineered gRNA in bacteriaReviewers' Comments:

Reviewer #1:

Remarks to the Author:

This study developed a synthetic tunable translation-level CRISPR interference (TI-CRISPRi) system by engineering guide RNAs and Cas13, which enabled precise and predictable control of mRNA translation. In detail, the author optimized the TI-CRISPRi system, evaluated its application in regulation of multiple genes and applied it in metabolic engineering for 3-HP production. This system provides a feasible approach for regulation of protein expression and possible strategy for pathway regulation. The manuscript is well organized and can be published in Nature Communications after addressing several concerns listed as below:

1. It is interesting that only dRfxCas13d functional well dRfxCas13d in downregulating the expression level of the reporter genes, while other orthologs had no function. Were there some correlations between the function and the enzyme structure? Suggest the authors give some possible explanation.
2. In figure 1b, figure S1C and else, the growth effect of TI-CRISPRi and different conditions. the measurement of final OD600 is not enough. Suggest the authors quantifying the growth rate μ_{max} and lag time, which can clearly show the growth profile of different strains.
3. In Fig. 4, engineering gRNA for tuning the TI-CRISPRi, the authors engineered several gRNA mutations for tuning gene expression. Is it possible to construct some model to predict the expression levels?

Suggest adding the expression level of mCherry w/o gRNA as a control in Figure 4C and Fig. S7a. The org. DR data can also be added in Fig. S7a.

4. In the part of 3-HP production (Figure 5), TI-CRISPRi based downregulation of competing pathways improved 3-HP production in batch-fermentation. How about the performance of this strategy in fed-batch fermentation? Suggesting adding experiments to compare the best regulated strains (such as strain aceE-fabI of figure 5c, S10 strain of Figure 5d) with the control strain under fed-batch fermentation (you can do fed-batch in shake-flask is OK). This would be helpful for future industrial application.

I noticed that only 8 g/L glucose was used for 3-HP production? Why use this low concentration of glucose, not the normal of 20 g/L glucose? What is working volume and what is the sample taking time? Please add the detailed information in material and methods, or figure legends.

Reviewer #2:

Remarks to the Author:

The manuscript from Kin et al entitled Tunable control of translation level by CRISPR-dCas13 in bacteria describe an optimised tool to repress translation in *E. coli* and *V. natrigens*. The work has clearly a lot of work behind as it can be seen in the many optimisation parameters that have been tuned to make the system efficient.

The manuscript is also timely and shows advantages over other expression level modulator systems, for example when tuning operons. The authors also demonstrated how the technology can be used for bioproduction.

While I think the manuscript is of high quality and interest for the scientific community, there are a few things I would like the authors to clarify before publication.

1. Across the manuscript, there are a lot of references to translational control (even in the title and abstract), but in reality, the work only explores gene downregulation. I suggest being more accurate in the wording across the manuscript to avoid confusion. In the same lines, the tool works in 2 gram negative bacteria but most reference is that the tool is universal for bacteria, do the authors think the system will work well in gram positive bacteriat too?
2. Three of the tested dCas13 did not work, why the authors think this is the case? were the enzymes expressed at all?

3. Experiment of figure 1b is missing a control with a gRNA targeting nothing or with no gRNA at all, to show that the effect is not due to the presence of the inducers or else.
4. Experiment of figure 1e, was the combination of GFP and nanoluc alone tested? this one seems to be missing for a systematic analysis.
5. lines 146- 151 clarify if the inconsistency in mRNA levels does somehow influence translation levels too. if not how does the cell compensate that fluctuation at mRNA levels?
6. line 173 "it was less" please say how much less.
7. Results in paragraph starting line 179, are there any correlations of RNA levels with the polar effect observed?
8. In the discussion: describe the limitations of the systems and further improvement. Discuss how activation can be achieved and combined as it has been done in other systems.

Reviewer #3:

Remarks to the Author:

In this work, the authors mutagenized dCas13 from four organisms, arriving upon dRfxCas1d, which is capable of excellent translational interference without hampering cellular growth. The author then characterizes this enzyme in detail but exploring plasmid downregulation of fluorescent proteins, chromosomal downregulation of integrated fluorescent proteins, the polar effect, and the potential downstream effects at target site milieu (i.e., downregulation in polycistronic reporters or essential genes). Spacers (i.e., guide RNA or gRNA sequences) were also explored by altering their sequence and length to maximize downregulation of a specific target with a juxtaposition to conventional dCas9 downregulation. Finally, systematic mutation of the direct repeat sequences in the guide RNA handle yielded remarkable tunability in gene expression of mCherry with demonstrable control between <5% and about 90% expression with respect to wild type. Finally, the TI-CRISPRi is applied to a complex, 48 gene circuit with a demonstrable improvement in 3-HB titer.

In general, the work is outstanding and well written. It has significant potential for engineering microbes outside the conventional boundaries of DNA-based dCas9. I have minor comments with regards to the detail and discussion that the authors provide in the text.

Specifically, I would like to see a better description of how the gRNAs/spacers were designed, a description of platform drawbacks (off-target effects, general strategy, etc) a mitigation of some of the broader claims made in the text, and a more thorough description of how the work could be applied in automation and machine learning pipelines, which are ultimately the future of metabolic engineering. Lastly, it may be worthwhile to investigate more DR variants in Figure 4E and Figure 5D to confirm whether knockdown strength indeed correlates between organisms and whether 3-HP production is so sensitive to precise knockdown, respectively.

In addition, please see specific and minor comments below.

Specific Comments:

Lines 79-83: What specific bacteria species are these enzymes from? You provide the phylogenetic tree in the supplement, but neither description provides their source. Likewise, those with less knowledge about dCas13 will probably not know the difference between dCas13a, dCas13b, etc. More detail would be useful here.

Lines 146-156: The nature of dCas13 means that mRNA measurement would be more meaningful than simply measuring reporter expression, though I appreciate the general difficulty described and the attempt the author's make here.

Figure 4d, 4e: Plotting the slope = 1 line here is misleading as it implies some sort of linear regression, however the caption notes that "r" represents the Spearman's correlation coefficient. Please replace "r" with the symbol for "rho" in the plots for clarity. A Spearman's rank correlation coefficient does not necessitate that the relationship is linear.

Lines 244-245: This claim, specifically the "universally" statement is far too broad. The authors have indeed demonstrated good tunability of their dCas13 in E. coli. Then, they show that seven out of fifty

of those DR candidates have expression in *Vibrio natriegens* correlating in rank with that of *E. coli*. The authors did not explore tunability outside of mCherry, nor did they explore more DRs in *Vibrio natriegens*. While the translation of the platform to *Vibrio natriegens* is encouraging, the authors would have to demonstrate tuning of many other genes in many other organisms to support this claim. The statement at the end of this paragraph, Lines 252-253, is much more sensible.

Lines 264-265: This section needs significant detail describing experiment design. Unless I'm much mistaken, the author's do not describe what DR(s) were selected. Were all 48 gene targets selected for high-level downregulation? 50% downregulation? Were all the initial DR sequences identical, then narrowed down based on promising targets?

Figure 5d: The jump in titer between S22 and S10 is remarkable, especially given that the relative increase in downregulation is around 5%. Is it possible that a weaker mutated DR could increase the likelihood of off target effects? We are operating under the assumption that DR downregulation efficiency for mCherry translates directly to *fabI*, which may not be the case.

Lines 343-354: The authors should provide some description about how a pipeline for automated assembly of gRNAs and various mutant DRs could be used with machine learning to generate optimized circuits of complex pathways. It would also be useful to describe how the platform could be multiplexed for downstream usage (e.g., target multiple pathways simultaneous with different strength of downregulation).

Lines 408-409: How were these spacers selected? What tools and criteria were used? How were off-target effects avoided? These must be addressed here and, briefly, in the main text. The authors didn't describe the potential for off-target effects anywhere in the text and this should most definitely be discussed somewhere.

Minor Comments:

Line 36: Rephrase "better" is a weak descriptor

Line 38: Remove "the" before gene expression

Line 79: "plenty" is a weak descriptor

Lines 330-332: Condense: "... bacterial cells concealed in Tx-CRISPRi-based screening"

Response to Reviewers' Comments

We appreciate the reviewers for providing us with fruitful comments. We believe the manuscript is now improved based on additional experiments and modifications as suggested by the reviewers. Here are the lists of experiments that we have conducted based on the comments.

1. Western blotting and Coomassie blue staining of protein gels to analyze the expression of four different dCas13 orthologs (Supplementary Fig. 1e and 1f) (by Comment 1-1 and 1-2)
2. Measuring specific growth rate of engineered bacterial strains (Fig. 1b, 3c, 3d, 5c, 5d and Supplementary Fig 1c) (by Comment 1-2)
3. Flask-level fed-batch fermentations of best-regulated strains for producing 3-HP (Supplementary Fig. 11) (by Comment 1-4)
4. Measuring mCherry fluorescence, specific growth rate, and optical cell density of strain with non-target guide RNA as a control while varying the concentration of aTc (Fig. 1b) (by Comment 2-3)
5. Additional combinations of guide RNAs for simultaneous knockdown of reporter genes (Fig. 1e) (by Comment 2-4)
6. Quantification of RNA-level perturbation caused by the TI-CRISPRi of the endogenous genes in *E. coli* (Supplementary Fig. 6a) (by Comment 2-7)
7. Adopting additional DR candidates for the fine-tuning of mCherry expression in *Vibrio natriegens* (Fig. 4d) (by Comment 3-4)
8. Adopting additional DR candidates and a spacer for the fine-tuning of *fabI* expression (Fig. 5d) (by Comment 3-6)

Following are the detailed responses to the reviewers' comments.

Reviewer #1

This study developed a synthetic tunable translation-level CRISPR interference (TI-CRISPRi) system by engineering guide RNAs and Cas13, which enabled precise and predictable control of mRNA translation. In detail, the author optimized the TI-CRISPRi system, evaluated its application in regulation of multiple genes and applied it in metabolic engineering for 3-HP production. This system provides a feasible approach for regulation of protein expression and possible strategy for pathway regulation. The manuscript is well organized and can be published in Nature Communications after addressing several concerns listed as below:

(Comment 1-1). It is interesting that only dRfxCas13d functional well dRfxCas13d in downregulating the expression level of the reporter genes, while other orthologs had no function. Were there some correlations between the function and the enzyme structure? Suggest the authors give some possible explanation.

(Response 1-1) We appreciate the reviewer's comment. Since the ternary complex of each Cas13 (Cas13 + guide RNA + target RNA) is mostly absent, we thought it would be hard to predict the relationship between the enzyme structure and its knockdown function. Instead, we investigated whether each dCas13 is functionally expressed as a soluble protein when the guide RNA and the target RNA are expressed together. We added His-tag(6x) at the N-terminal region of dCas13 orthologs to specifically detect them with the western blot. Indeed, only the dRfxCas13d was functionally expressed as a soluble form, which showed one clear band and similar band thickness between total and soluble fraction in the His-tag specific western blot. However, other dCas13 proteins, which could not repress the translation of the reporter mRNA, were not well detected in the soluble fraction. This result demonstrates the importance of the functional expression of the dCas13 protein to work as a robust translation-level repressor.

This is described in the revised manuscript as follows: *“When we investigated the expression of each dCas13 ortholog by western blotting and coomassie blue staining, only the dRfxCas13d was clearly detected in both the total and soluble fraction of bacterial cells (Supplementary Fig. 1e and 1f). This result emphasizes the appropriate choice of dCas13 ortholog to be expressed in the functional form for efficient gene knockdown.”* (Page 5, line 95-99 in the revised version)

Supplementary Figure 1e and 1f. For the strains expressing mCherry-targeting guide RNA, the expression of dCas13 with N-terminal His-tag(6x) was analyzed by the western blot (e) and the coomassie blue staining (f). The location of the full-length protein band was denoted by an asterisk (dLb-NHis: 140.3kDa, dBz-NHis: 147.6kDa, dPb-NHis: 135.6kDa, dRfx-NHis: 113.6kDa). control, parental strain without dCas13 expression; dLb, dLbuCas13a; dBz, dBzCas13b; dPb, dPbCas13b; dRfx, dRfxCas13d; T, total fraction; S, soluble fraction.

(Comment 1-2). In figure 1b, figure S1C and else, the growth effect of TI-CRISPRi and different conditions. the measurement of final OD₆₀₀ is not enough. Suggest the authors quantifying the growth rate μ_{max} and lag time, which can clearly show the growth profile of different strains.

(Response 1-2) We thank the reviewer for giving us a constructive suggestion. The experiments of Figure 1b, S1c, 3c & 3d, 5c & 5d showed the difference in bacterial growth only in terms of final OD₆₀₀. Therefore, to investigate the cellular growth of these strains during the exponential phase, we measured their maximum specific growth rate, which is depicted in each figure.

For most cases, the lag time after refreshing culture media and the induction of dCas13 was trivial or not observed. Also, the maximum specific growth rate determined in the mid-log phase (by the slope of $\ln(\text{OD}_{600})$ vs. time) was closely correlated to the final OD₆₀₀ in most experiments.

This is described in the revised manuscript as follows: “For measuring the maximum specific growth rate of *E. coli* under different conditions, the overnight culture of each colony was refreshed into a ratio of 1:100 with 100 ng/ml of aTc and loaded on the 96-well microplate, which was incubated at 37°C with 600 rpm shaking using a Hidex. The maximum specific growth rate was quantified from linear regression of logarithmic OD₆₀₀ during the exponential phase. The OD₆₀₀ data of time points between 1 to 3 hours were used to calculate specific growth rates, except for the experiment of Figure 5, which used the data of 2 to 4 hours.” (Page 24, line 492-498 in the revised version)

Here are the figures that we have updated with newly measured specific growth rates:

Figure 1b. Input-responsive characteristics of the TI-CRISPRi system. The dCas13 gene was expressed under the aTc-inducible promoter (P_{tet}). The relative expression level of mCherry was evaluated by normalizing the RFU/OD₆₀₀ of mCherry without aTc as 100%.

Supplementary Figure 1c. Comparison of four different dCas13 orthologs as a synthetic gene expression modulator for the TI-CRISPRi system. The expression level of mCherry (b), the specific growth rate of exponential phase (c), and the final cellular density (d) of bacterial strains were measured.

Figure 3c & 3d. Diminished polar effect and growth retardation by the TI-CRISPRi while targeting the endogenous operons of *Escherichia coli*. The cellular growth while performing Tx-CRISPRi (c) or TI-CRISPRi (d) toward the upstream non-essential genes was compared with the strain harboring the non-target guide RNA (NT).

Figure 5c & 5d. Metabolic pathway redirection via tunable TI-CRISPRi for enhancing the production of 3-HP. (c) The TI-CRISPRi simultaneously knocked down two genes from the five selected candidates in (b). (d) To fine-tune the expression level of the *fabI* gene, 16 different guide RNAs consisting of eight different DR sequences and the two *fabI* spacer (*spv1* and *spv2*) were adopted. The 3-HP titer and the OD₆₀₀ were measured after 24 hours of the induction of the *mcr* gene.

(Comment 1-3) In Fig. 4, engineering gRNA for tuning the TI-CRISPRi, the authors engineered several gRNA mutations for tuning gene expression. Is it possible to construct some model to predict the expression levels?

Suggest adding the expression level of mCherry w/o gRNA as a control in Figure 4C and Fig. S7a. The org. DR data can also be added in Fig. S7a.

(Response 1-3) We appreciate the reviewer's comment and tried to fit the expression level of mCherry and the minimum free energy of DR folding for the stem-mutated DRs. Unfortunately, there was no remarkable correlation between DR's folding energy and the resulting mCherry expression level. This result implies that only the numerical estimation of DR could not accurately predict the repression strength or the rank of DR, which necessitates the experimental data of DR in each biological system.

This is described in the revised manuscript as follows: "Indeed, the minimum folding energy of each stem-mutated DR and the relative expression level of mCherry derived from DR-mutated guide RNAs did not show remarkable correlation (Supplementary Fig.

8)." (Page 12, line 244-246 in the revised version)

Supplementary Figure 8. Comparison of relative expression level of mCherry derived from stem-mutated guide RNAs and minimum free energy of each stem-mutated DR. The minimum free energy (b, c) of the modified direct repeat was calculated based on the NUPACK software¹. All of the relative expression data and the error bars were from Supplementary Figure 7.

Also, we added the normalized expression level of mCherry for the strain with the non-target guide RNA as a control in Figure 4c and Supplementary Figure 7a. The org. DR data was also added in Supplementary Fig. 7a. We thank the reviewer for pointing out them.

Figure 4c. Development and application of tunable TI-CRISPRi based on the handle-engineered guide RNAs. (c) The resulting expression levels of mCherry derived from the guide RNAs harboring diverse DR sequences are shown in the graph.

Supplementary Fig.7a. Relative expression level of mCherry derived from stem-mutated guide RNAs. The 6-bp stem structure of the direct repeat was mutated for three different objectives: (a) Disrupting the base pairing, (b) Replacing the GC pair into the AU pair, and (c) Extending or shortening the stem length.

(Comment 1-4) In the part of 3-HP production (Figure 5), TI-CRISPRi based downregulation of competing pathways improved 3-HP production in batch-fermentation. How about the performance of this strategy in fed-batch fermentation? Suggesting adding experiments to compare the best regulated strains (such as strain *aceE-fabI* of figure 5c, S10 strain of Figure 5d) with the control strain under fed-batch fermentation (you can do fed-batch in shake-flask is OK). This would be helpful for future industrial application. I noticed that only 8 g/L glucose was used for 3-HP production? Why use this low concentration of glucose, not the normal of 20 g/L glucose? What is working volume and what is the sample taking time? Please add the detailed information in material and methods, or figure legends.

(Response 1-4) We appreciate the reviewer's comment. During revision, we conducted a flask-level fed-batch experiment on the NT strain, *aceE-fabI* strain of Figure 5c, and S10 – *fabI* spv1 strain of Figure 5d, starting on 20 g/L glucose. We could observe the remarkable enhancement of 3-HP titer by TI-CRISPRi also in the flask scale, 20g/L of glucose, and the fed-batch culture.

The reason why we conducted usually on the 8 g/L of glucose was to facilitate the screening process, since the *E. coli* entirely consumed this concentration of glucose in 24 hours after refreshing the culture media. Although we utilized this low concentration of glucose, this experimental condition yielded a range of 3-HP titers depending on the repressed gene targets of TI-CRISPRi. Also, for the best-regulated strains, the inclination

of 3-HP titer enhancement was maintained even if the culture volume and the initiation concentration of glucose were increased, as we could observe in the fed-batch experiment. For all the experiments in Figure 5, we worked on a 2 mL volume in the test tube culture, and the sampling time was 24 hours after inducing the expression of *mcr* by IPTG, which is now described in the method section.

This is described in the revised manuscript as follows: “Since all of these TI-CRISPRi experiments on optimizing 3-HP production were conducted based on the small-scale test tube culture with a small amount of glucose (8 g/L), we investigated whether the effect of the TI-CRISPRi repression could be maintained for a larger scale of culture and a higher concentration of carbon source. We performed flask-scale fed-batch culture using the best-regulated strains from the TI-CRISPRi screening, starting from 20 g/L of glucose. We observed that the 3-HP titers were also remarkably increased in these strains compared to the non-targeted strain, which was 16.4-fold higher in strain with S10-*fabI* *spv1* guide RNA (Supplementary Fig. 11). This result suggests that our tunable TI-CRISPRi could also robustly function and redirect the native metabolic flux of microbial cell factories.” (Page 16, line 329-337 in the revised version)

“For the experiments in Figure 5, we conducted the bacterial culture in the test tube with a 2 mL volume of media. The overnight culture was diluted at an OD_{600} of 0.05 and induced with 100 ng/ml of aTc for the expression of *dCas13*. Subsequently, when the OD_{600} of the culture was reached at 0.8, 20 μ M of IPTG was added to induce the *mcr* gene. After 24 hours of shaking incubation, the culture broth was centrifuged, and the supernatant was sampled to measure the concentration of 3-HP. The fed-batch fermentations in Supplementary Figure 11 were performed in the 25 mL of culture volume in the 250 mL baffled flask and a 20 g/L initial concentration of glucose. The overall scheme of fermentation was the same with the test tube culture except for the concentration of IPTG, which was 200 μ M in the fed-batch fermentation.” (Page 25, line 565-573 in the revised version)

Supplementary Figure 11. Fed-batch fermentation profile of strains under the TI-CRISPRi regulation. (a) the strain with non-target guide RNA. (b) the strain with two guide RNAs each targeting *aceE* and *fabI*. (c) the strain repressing *fabI* with the mutated guide RNA (S10-*fabI* *spv1*). Circle, OD_{600} ; rectangle, glucose; diamond, 3-HP; triangle, acetate; reverse-triangle, pyruvate.

Reviewer #2

The manuscript from Kin et al entitled Tunable control of translation level by CRISPR-dCas13 in bacteria describe an optimised tool to repress translation in *E. coli* and *V. natrigens*. The work has clearly a lot of work behind as it can be seen in the many optimisation parameters that have been tuned to make the system efficient.

The manuscript is also timely and shows advantages over other expression level modulator systems, for example when tuning operons. The authors also demonstrated how the technology can be used for bioproduction.

While I think the manuscript is of high quality and interest for the scientific community, there are a few things I would like the authors to clarify before publication.

(Comment 2-1) Across the manuscript, there are a lot of references to translational control (even in the title and abstract), but in reality, the work only explores gene downregulation. I suggest being more accurate in the wording across the manuscript to avoid confusion. In the same lines, the tool works in 2 gram negative bacteria but most reference is that the tool is universal for bacteria, do the authors think the system will work well in gram positive bacteria too?

(Response 2-1) We appreciate the reviewer's comment, which points out that some wordings (e.g. "control", "universal") in this manuscript should be corrected for clarity. Across the manuscript, we changed these wordings into more specific ones to avoid confusion. Since the word "control" includes both up- and down-regulation and our work only focused on the repression of mRNA translation, they were substituted into "down-regulate", "suppress", or "repress". The word "universal" is used in the line 252, which was intended to highlight the tunability of TI-CRISPRi across other genes in *E. coli*. However, we thought this wording was a quite broad claim and should be mitigated. Therefore, we also deleted this wording and toned down other broad expressions into more specific ones. In line with this response, we changed the title as "Tunable translation-level CRISPR interference by dCas13 and engineered gRNA in bacteria".

Also, we think this system could be potentially applied to gram-positive bacteria with adequate characterization and optimization in each organism, as the dCas9-based Tx-CRISPRi was expanded to various non-model bacteria. We added this opinion on the expandability of our tool in the discussion part to give some prospects to the readers of this article.

This is described in the revised manuscript as follows:

"These results validate that our tunable TI-CRISPRi system could be adopted to fine-tune the expression level of various genes in E. coli." (Page 12-13, line 259-260 in the revised version)

"While we applied the tunable TI-CRISPRi in two gram-negative bacterial strains, detailed characterization on bacterial strains other than E. coli should proceed to validate the expandability of our system toward diverse strains. We believe that further investigation

and optimization could enable the usage of this tool in diverse bacterial systems, including gram-positive bacteria, as other CRISPR-based tools have been successfully adopted and widely utilized in diverse strains^{71,72}.” (Page 18, line 380-384)

(Comment 2-2). Three of the tested dCas13 did not work, why the authors think this is the case? were the enzymes expressed at all?

(Response 2-2) We appreciate the reviewer’s comment. According to this comment, we investigated whether each dCas13 is functionally expressed as a soluble protein when the guide RNA and the target RNA are expressed together. We added His-tag(6x) at the N-terminal region of dCas13 orthologs to specifically detect them with the western blot. Indeed, only the dRfxCas13d was functionally expressed as a soluble form, which showed one clear band and similar band thickness between total and soluble fraction in the His-tag specific western blot. However, other dCas13 proteins, which could not repress the translation of the reporter mRNA, were not well detected in the soluble fraction. This result demonstrates the importance of the functional expression of the dCas13 protein to work as a robust translation-level repressor.

This is described in the revised manuscript as follows: “When we investigated the expression of each dCas13 ortholog by western blotting and coomassie blue staining, only the dRfxCas13d was clearly detected in both the total and soluble fraction of bacterial cells (Supplementary Fig. 1e and 1f). This result emphasizes the appropriate choice of dCas13 ortholog to be expressed in the functional form for efficient gene knockdown.” (PAGE 5, line 95-99 in the revised version)

Supplementary Figure 1e and 1f. For the strains expressing mCherry-targeting guide RNA, the expression of dCas13 with N-terminal His-tag(6x) was analyzed by the western blot (e) and the coomassie blue staining (f). The location of the full-length protein band was denoted by an asterisk (dLb-NHis: 140.3kDa, dBz-NHis: 147.6kDa, dPb-NHis: 135.6kDa, dRfx-NHis: 113.6kDa). control, parental strain without dCas13 expression; dLb, dLbuCas13a; dBz, dBzCas13b; dPb, dPbCas13b; dRfx, dRfxCas13d; T, total fraction; S, soluble fraction.

(Comment 2-3) Experiment of figure 1b is missing a control with a gRNA targeting nothing or with no gRNA at all, to show that the effect is not due to the presence of the inducers or else.

(Response 2-3) We are sorry for missing the control data from the strain with non-target guide RNA. We added the experimental data (mCherry fluorescence, specific growth rate, final OD₆₀₀) from that strain under diverse concentrations of aTc. The fluorescence data confirmed that the decrease of mCherry expression level due to adding aTc was far more significant in the T strain, expressing the guide RNA targeting mCherry mRNA, than the strain with non-targeting guide RNA.

This is described in the revised manuscript as follows: “Notably, the decrease in mCherry level upon the addition of aTc was more remarkable when the mCherry-targeting guide RNA was expressed, compared to the non-target strain (Fig. 1b).” (Page 6, line 105-107 in the revised version)

Figure 1b. Input-responsive characteristics of the TI-CRISPRi system. The dCas13 gene was expressed under the aTc-inducible promoter (P_{tet}). The relative expression level of mCherry was evaluated by normalizing the RFU/OD₆₀₀ of mCherry without aTc as 100(%).

(Comment 2-4) Experiment of figure 1e, was the combination of GFP and nanoluc alone tested? this one seems to be missing for a systematic analysis.

(Response 2-4) We are sorry for missing the combination of mCherry & nanoluc and GFP & nanoluc in this experiment. During revision, we tested those combinations, and they also yielded specific knockdown of targeted genes without decreasing the expression level of non-targeted gene. Here is the figure that we have updated following this comment:

Figure 1e. Simultaneous and specific reporter gene knockdown by the TI-CRISPRi. Each guide RNA with the effective spacer (mCherry sp1, GFP sp1, nanoluc sp1) specifically down-regulated the target gene. Introducing multiple guide RNAs knocked down all of the targeted genes.

(Comment 2-5) lines 146- 151 clarify if the inconsistency in mRNA levels does somehow influence translation levels too. if not how does the cell compensate that fluctuation at mRNA levels?

(Response 2-5) We are sorry for the confusing statement in this paragraph. Highlighting the inconsistency in the decrease of reporter mRNA seemed unimportant in the context of the entire manuscript. To clarify, we deleted the statement that the decrease in target mRNA was inconsistent. Instead, we emphasized that some extent of mRNA degradation occurred by the TI-CRISPRi, although the dCas13 does not directly block transcription by targeting the double-stranded DNA or recruiting endogenous RNase complex. We speculated that the degradation of mRNA caused by translation inhibition is presumably due to the decreased shielding effect of the ribosomes on mRNA which blocks the access of RNase and Rho terminator to the actively translated mRNA.

This is described in the revised manuscript as follows: “We next investigated whether the amount of targeted mRNA could also be affected by TI-CRISPRi. Because of the co-localized and coordinated transcription/translation processes in a prokaryotic cell, suppressing the translation of the mRNA could lead to its degradation and premature transcription termination due to the absence of ribosomes protecting the mRNA³⁰. We found that the TI-CRISPRi decreased the quantity of all three reporter mRNA, although it was less significant than the decrease in protein level (Supplementary Fig. 4). This result suggests that excluding the ribosome by blocking translation initiation via TI-CRISPRi could moderately decrease the amount of targeted mRNA, even though the dCas13 lacks the function of blocking transcription by targeting the double-stranded DNA^{9,15} or recruiting endogenous RNases as sRNA-Hfq does^{31,32}”. (Page 8, line 153-162 in the revised version)

(Comment 2-6) line 173 "it was less" please say how much less.

(Response 2-6) We are sorry for the insufficient description of the polar effect in this line. However, prior to this sentence, we quantitatively mentioned the less significant polar effect by the TI-CRISPRi in terms of a decrease in the downstream GFP expression level while comparing with the Tx-CRISPRi (Line 171; When the upstream mCherry was targeted, the Tx-CRISPRi led to far reduced expression of the downstream GFP (19.6%, Fig 2c) compared to the TI-CRISPRi (63.1%, Fig 2d).) Since the description after line 177 mainly focused on discussing why some degree of polar effect was also observed in the TI-CRISPRi, we removed the phrase “although it was less than Tx-CRISPRi” for clarification.

This is described in the revised manuscript as follows: “*However, it should be noted that the TI-CRISPRi also showed some degree of polar effect.*” (Page 9, line 178-179 in the revised version)

(Comment 2-7) Results in paragraph starting line 179, are there any correlations of RNA levels with the polar effect observed?

(Response 2-7) We appreciate the reviewer’s comment. The polar effect derived from the Tx-CRISPRi indicates the co-suppression of downstream gene expression located in the same operon, although the upstream gene was targeted. This phenomenon arises from the nature of the Tx-CRISPRi, which roadblocks transcription and directly reduces the amount of mRNA encoding the target gene or operon. For the TI-CRISPRi, we observed in the experiment of Figure 2 and Supplementary Figure 5 that the polar effect was far reduced compared to the Tx-CRISPRi. For the experiment in Figure 3, we expected that the decrease in mRNA level derived from the TI-CRISPRi would be moderate compared to the decline of protein expression, which was also observed in Supplementary Figure 4. Therefore, we performed an RT-qPCR experiment to measure the amount of the targeted non-essential endogenous genes. We observed that the mRNA decreased to about 40~70% compared to the non-target strain, which was less significant than the decline of the protein expression level (decreased to <12% compared to the non-target strain). This result implies that the moderate or slight decrease of RNA level from the TI-CRISPRi contributes to the less significant polar effect than the Tx-CRISPRi.

This is described in the revised manuscript as follows: “*When we quantified the mRNA level of these genes, it was observed that the TI-CRISPRi induced only a moderate or weak decrease of mRNA level, which ranges from 40% to 70% compared to the non-target control strain (Supplementary Fig. 6a). We speculated that this moderate decrease of mRNA contributed to a less significant polar effect in contrast to the Tx-CRISPRi.*” (Page 10, line 195-199 in the revised version)

Supplementary Fig. 6 Perturbation on the RNA- and protein-level caused by the TI-CRISPRi toward the endogenous genes in *E. coli*. (a) The mRNA level was quantified by RT-qPCR experiment to inspect whether the amount of targeted endogenous transcript was decreased by the TI-CRISPRi. (b) To investigate whether the TI-CRISPRi effectively knocked down the five endogenous genes targeted in the experiment of Figure 3, nanoluciferase (*nanoluc*) was fused to the C-terminal domain of each gene and the relative luminescence level of the strain with the effective guide RNA (T) was measured against that of the strain with non-target guide RNA (NT). Therefore, the relative luminescence level represents the decreased expression level of the gene targeted by TI-CRISPRi. The error bar represents the standard deviation from the biologically independent cell cultures ($n = 3$), and the white dots indicate the actual data points. The *P*-value of each strain's dataset was determined by the two-tailed Student's *t*-test compared to the dataset of the NT strain. The asterisk indicates the *P*-value. * $P < 0.05$, ** $P < 0.01$, *** $P < 0.001$.

(Comment 2-8) In the discussion: describe the limitations of the systems and further improvement. Discuss how activation can be achieved and combined as it has been done in other systems.

(Response 2-8) We thank the reviewer's comment for expanding the discussion, which could give more prospects to the readers of this article. As we thought, the limitations of this research could be narrowed down into three points: 1) The analysis on the possible off-target effect occurred by the TI-CRISPRi is absent, which was due to the difficulties in

performing proteome analysis to investigate the genome-wide translational fluctuations; 2) The validation of tunable TI-CRISPRi system in the gram-positive bacteria is absent; 3) We only focused on the fine-tuned gene repression, which did not include the inspection on the gene activation.

However, we think these limitations suggest valuable points of investigation and further research. Although it is hard to analyze the proteome, analyzing the possible off-targets from the TI-CRISPRi could suggest possible design rules of spacer sequences to maximize the specificity toward target genes. For the gram-positive bacteria, as if the Tx-CRISPRi using the dCas9 or dCpf1 was further expanded from the model strain *E. coli* toward diverse non-model bacteria, we believe that the TI-CRISPRi could also be harnessed for the several bacterial systems accompanied by systematic optimization of genetic parts since this tool does not rely on the endogenous components like Hfq-sRNA. For the targeted activation of mRNA translation via the dCas13, one previous study revealed that fusing the native translation initiation factor of *E. coli* led to the targeted activation of mRNA. We anticipate that applying the mutant DR sequences in this translational activation could yield diverse gene activation magnitudes.

Otoupal, P. B., Cress, B. F., Doudna, J. A. & Schoeniger, J. S. CRISPR-RNAa: targeted activation of translation using dCas13 fusions to translation initiation factors. *Nucleic Acids Res.* 1–13 (2022)

This is described in the revised manuscript as follows:

“We observed that only the reporter genes targeted by the guide RNAs were specifically knocked down in multiplexed gene targeting, which suggests the high specificity of our system. However, due to the difficulties in performing proteome analysis, we could not inspect the genome-wide off-target perturbations of translational inhibition under the application of TI-CRISPRi. Further investigation on proteome-level perturbation caused by TI-CRISPRi could suggest a detailed understanding of the off-target effect and propose design rules of guide RNA to enhance specificity.” (Page 16-17, line 349-355 in the revised version)

“Though we focused on the down-regulation of gene expression level, our attenuated guide RNAs might be applied to achieve diverse magnitudes of gene activation if the dCas13 is fused with translational activators¹³.” (Page 17, line 366-368 in the revised version)

*“While we applied the tunable TI-CRISPRi in two gram-negative bacterial strains, detailed characterization on bacterial strains other than *E. coli* should proceed to validate the expandability of our system toward diverse strains. We believe that further investigation and optimization could enable the usage of this tool in diverse bacterial systems, including gram-positive bacteria, as other CRISPR-based tools have been successfully adopted and widely utilized in diverse strains^{71,72}.”* (Page 18, line 380-384 in the revised version)

Reviewer #3

In this work, the authors mutagenized dCas13 from four organisms, arriving upon dRfxCas1d, which is capable of excellent translational interference without hampering cellular growth. The author then characterizes this enzyme in detail but exploring plasmid downregulation of fluorescent proteins, chromosomal downregulation of integrated fluorescent proteins, the polar effect, and the potential downstream effects at target site milieu (i.e., downregulation in polycistronic reporters or essential genes). Spacers (i.e., guide RNA or gRNA sequences) were also explored by altering their sequence and length to maximize downregulation of a specific target with a juxtaposition to conventional dCas9 downregulation. Finally, systematic mutation of the direct repeat sequences in the guide RNA handle yielded remarkable tunability in gene expression of mCherry with demonstrable control between <5% and about 90% expression with respect to wild type. Finally, the TI-CRISPRi is applied to a complex, 48 gene circuit with a demonstrable improvement in 3-HB titer. In general, the work is outstanding and well written. It has significant potential for engineering microbes outside the conventional boundaries of DNA-based dCas9. I have minor comments with regards to the detail and discussion that the authors provide in the text. Specifically, I would like to see a better description of how the gRNAs/spacers were designed, a description of platform drawbacks (off-target effects, general strategy, etc) a mitigation of some of the broader claims made in the text, and a more thorough description of how the work could be applied in automation and machine learning pipelines, which are ultimately the future of metabolic engineering. Lastly, it may be worthwhile to investigate more DR variants in Figure 4E and Figure 5D to confirm whether knockdown strength indeed correlates between organisms and whether 3-HP production is so sensitive to precise knockdown, respectively.

(Response) The comments described above were repeated in the following specific comments. Thus, we wrote our detailed responses under each specific comment.

(Comment 3-1) Lines 79-83: What specific bacteria species are these enzymes from? You provide the phylogenetic tree in the supplement, but neither description provides their source. Likewise, those with less knowledge about dCas13 will probably not know the difference between dCas13a, dCas13b, etc. More detail would be useful here.

(Response 3-1) We thank the reviewer for the suggestion and added each name of the bacterial species which are the sources of these enzymes in the Supplementary Figure 1a. Also, we added some description about the classification of Cas13a ~ d, which is based on the presence of accessory proteins and the primary sequence of each Cas13 protein.

This is described in the revised manuscript as follows: *“Several previous studies have discovered hundreds of Cas13 orthologs that can bind to target RNAs across a wide range of bacterial species with the help of genome mining and classified into four subtypes (Cas13a, Cas13b, Cas13c, and Cas13d) depending on the existence of accessory proteins or the primary amino acid sequences of the Cas13 effectors¹⁴⁻¹⁷.”* (Page 5, line 79-82 in the revised version)

Supplementary Figure 1a. Comparison of four different dCas13 orthologs as a synthetic gene expression modulator for the TI-CRISPRi system. (a) Phylogenetic tree of Cas13 orthologs and the selected Cas effector proteins (left). For the inactivation of non-specific nuclease activity of Cas13, the main catalytic residues of the HEPN domain (RxxxxH) were identified and substituted to the alanine (right), creating the dead Cas13 effector. The bacterial source of each Cas13 ortholog was denoted in the parenthesis.

(Comment 3-2) Lines 146-156: The nature of dCas13 means that mRNA measurement would be more meaningful than simply measuring reporter expression, though I appreciate the general difficulty described and the attempt the author's make here.

(Response 3-2) We are sorry for the confusing statement in this paragraph. We attempted to highlight that blocking translation via TI-CRISPRi could also lead to mRNA degradation due to the coupling between transcription and translation. However, the decrease in mRNA level was less severe than that of protein level for all three reporter genes. Therefore, we thought mRNA measurement in TI-CRISPRi would give us some insight into this transcription-translation coupling phenomenon. As the TI-CRISPRi directly blocks the translation of mRNA, the protein measurement is sufficient for evaluating the nature of TI-CRISPRi. We rewrote this paragraph to make this message clear.

This is described in the revised manuscript as follows: "We next investigated whether the amount of targeted mRNA could also be affected by TI-CRISPRi. Because of the co-localized and coordinated transcription/translation processes in a prokaryotic cell, suppressing the translation of the mRNA could lead to its degradation and premature transcription termination due to the absence of ribosomes protecting the mRNA³⁰. We found that the TI-CRISPRi decreased the quantity of all three reporter mRNA, although it was less significant than the decrease in protein level (Supplementary Fig. 4). This result suggests that excluding the ribosome by blocking translation initiation via TI-CRISPRi could moderately decrease the amount of targeted mRNA, even though the dCas13 lacks the function of blocking transcription by targeting the double-stranded DNA^{9,15} or recruiting endogenous RNases as sRNA-Hfq does^{31,32}." (Page 8, Line 153-162 in the revised version)

(**Comment 3-3**) Figure 4d, 4e: Plotting the slope = 1 line here is misleading as it implies some sort of linear regression, however the caption notes that “r” represents the Spearman’s correlation coefficient. Please replace “r” with the symbol for “rho” in the plots for clarity. A Spearman’s rank correlation coefficient does not necessitate that the relationship is linear.

(**Response 3-3**) We are very sorry for the confusion. As the reviewer commented, we removed the slope = 1 line and denoted Spearman’s correlation coefficient as ρ (rho) in Figure 4d and 4e. Here are the figures that we have updated following this comment:

Figure 4d and 4e. Development and application of tunable TI-CRISPRi based on the handle-engineered guide RNAs. (d) Relative GFP and nanoluciferase expression level against the mCherry while using the guide RNA with the same mutated DR sequence. GFP sp2 and nanoluc sp1 were each adopted for the knockdown of GFP and nanoluciferase as a spacer sequence. All of the reporter genes were expressed from the chromosome of *E. coli*. (e) The relative expression level of mCherry in the *Vibrio natriegens* compared to that of *Escherichia coli* when the same attenuated guide RNAs were adopted. The Spearman’s correlation coefficient (ρ) and the related p -value are annotated in each graph (d, e).

(**Comment 3-4**) Lines 244-245: This claim, specifically the “universally” statement is far too broad. The authors have indeed demonstrated good tunability of their dCas13 in *E. coli*. Then, they show that seven out of fifty of those DR candidates have expression in *Vibrio natriegens* correlating in rank with that of *E. coli*. The authors did not explore tunability outside of mCherry, nor did they explore more DRs in *Vibrio natriegens*. While the translation of the platform to *Vibrio natriegens* is encouraging, the authors would have to demonstrate tuning of many other genes in many other organisms to support this claim. The statement at the end of this paragraph, Lines 252-253, is much more sensible.

(**Response 3-4**) We are sorry for the confusion. We used the word “universally” in line 244-245 to highlight the tunability of TI-CRISPRi in *E. coli*, not for the other untested organisms, as we observed in Figure 4d. However, this wording still would be a quite broad claim, which should be mitigated. Therefore, we deleted this wording and also toned down other broad expressions into more specific ones.

According to the reviewer's comment, we tested more mutant DR sequences to fine-tune the expression of mCherry in *Vibrio natriegens*. We observed that the positive correlation between these two strains was still maintained. However, we did not test whether tunability is maintained across many other genes and organisms besides *E. coli*. Therefore, as the reviewer pointed out, we would not insist that this tunable TI-CRISPRi is universal for all the other bacterial systems since it should be explored in more detail in further investigations. We discussed the necessity of further investigations to expand the scope of bacterial strains to adopt the TI-CRISPRi.

This is described in the revised manuscript as follows: “*These results validate that our tunable TI-CRISPRi system could be adopted to fine-tune the expression level of various genes in E. coli.*” (Page 12-13, line 259-260 in the revised version)

Figure 4e. Development and application of tunable TI-CRISPRi based on the handle-engineered guide RNAs. (e) The relative expression level of mCherry in the *Vibrio natriegens* compared to that of *Escherichia coli* when the same attenuated guide RNAs were adopted. The Spearman's correlation coefficient (ρ) and the related p-value are annotated in each graph (d, e).

“While we applied the tunable TI-CRISPRi in two gram-negative bacterial strains, detailed characterization on bacterial strains other than *E. coli* should proceed to validate the expandability of our system toward diverse strains. We believe that further investigation and optimization could enable the usage of this tool in diverse bacterial systems, including gram-positive bacteria, as other CRISPR-based tools have been successfully adopted and widely utilized in diverse strains^{71,72}.” (Page 18, line 380-384 in the revised version)

(Comment 3-5) Lines 264-265: This section needs significant detail describing experiment design. Unless I'm much mistaken, the author's do not describe what DR(s) were selected. Were all 48 gene targets selected for high-level downregulation? 50% downregulation? Were all the initial DR sequences identical, then narrowed down based on promising targets?

(Response 3-5) We are sorry for the lack of a detailed description. The initial gene screening in Figure 5b was performed based on the guide RNA with the original DR sequence to ensure high-level downregulation. Among the gene targets, the knockdown of *fabI* led to a high titer of 3-HP together with the growth retardation. Therefore, to balance the cellular growth and the 3-HP titer, we adopted the mutated DR sequences to fine-tune the expression level of *fabI*.

This is described in the revised manuscript as follows: “*We selected 48 genes and targeted them using the guide RNAs with the original DR sequence to ensure effective gene repression and determine which gene target knockdown could enhance the titer of 3-HP (Fig. 5a).*” (Page 13, line 279-281)

(Comment 3-6) Figure 5d: The jump in titer between S22 and S10 is remarkable, especially given that the relative increase in downregulation is around 5%. Is it possible that a weaker mutated DR could increase the likelihood of off target effects? We are operating under the assumption that DR downregulation efficiency for mCherry translates directly to *fabI*, which may not be the case.

(Response 3-6) We appreciate the reviewer's comment, which points out whether the 3-HP production is so sensitive to the various levels of *fabI* knockdown and whether the off-target effect could increase when the weaker mutated DR is used. To implement more diverse knockdown strengths on the *fabI* gene, we additionally tested one more spacer (*fabI* spv2) targeting more closely to the shine-dalgarno sequence than the original spacer (*fabI* spv1) and also adopted DR variants on it. We observed that the 3-HP titer suddenly dropped while attenuating the repression strength on *fabI* when we used this new spacer. This result demonstrates the sensitivity of 3-HP production while the fatty-acid synthesis cycle was repressed at various levels.

As for the relationship between the off-target effect and the weaker mutated DR, we would like to refer to our previous study on tunable Tx-CRISPRi. This system also attenuated the handle structure of sgRNA and showed that using the weaker sgRNA handle rather decreased the off-targets compared to the original sgRNA. Although we could not directly evaluate the off-target effect due to the inability to perform proteome analysis, we expect that utilizing our DR-mutated guide RNAs in the TI-CRISPRi might instead decrease the off-target effect.

Byun, G., Yang, J. & Seo, S. W. CRISPRi-mediated tunable control of gene expression level with engineered single-guide RNA in *Escherichia coli*. *Nucleic Acids Res.* 51, 4650–4659 (2023).

The reviewer also suggested that the downregulation efficiency of each DR observed in mCherry could not be directly translated to *fabI*. We do admit the reviewer's comment since in Figure 4d, the rank of DR while targeting GFP or nanoluciferase was not perfectly matched with the rank from mCherry. However, since there was a strong positive correlation between them, we expect that our mutated DR sequences could be utilized to predictably down-regulate other genes, as we also showed for the fine-tuned knockdown of *fabI* in Figure 5d.

This is described in the revised manuscript as follows: "To implement diverse knockdown strength on the *fabI*, we designed another spacer (*fabI* spv2) targeting more upstream of the *fabI* mRNA than the original spacer (*fabI* spv1) and adopted mutant DR sequences to both spacers, yielding 16 different guide RNAs to repress *fabI* in total. We found that the cellular growth gradually increased as the weaker DR sequence was utilized in both spacers (Fig. 5d). It was also noticeable that the optimal DR sequence reaching a maximal 3-HP production differed in these two spacers (S19 in *fabI* spv2 and S10 in *fabI* spv1). In total, the titer of 3-HP reached the maximum of 2.42 g/L when the guide RNA with S10 DR and *fabI* spv1 spacer was used, which was 14.2-fold improved against the strain with non-target guide RNA (Fig. 5d). However, the titer of 3-HP was suddenly dropped as the weaker DR sequence was adopted for both spacers. These results demonstrate that adjusting the expression level of *fabI* in a fine-tuned manner for securing an adequate amount of malonyl-CoA was critical for diminishing growth retardation and achieving the optimal production of 3-HP." (Page 15, line 313-325 in the revised version)

Figure 5d. Metabolic pathway redirection via tunable TI-CRISPRi for enhancing the production of 3-HP. (d) To fine-tune the expression level of the *fabI* gene, 16 different guide RNAs consisting of eight different DR sequences and the two *fabI* spacer (*spv1* and *spv2*) were adopted. The 3-HP titer and the OD₆₀₀ were measured after 24 hours of the induction of the *mcr* gene.

(Comment 3-7) Lines 343-354: The authors should provide some description about how a pipeline for automated assembly of gRNAs and various mutant DRs could be used with machine learning to generate optimized circuits of complex pathways. It would also be useful to describe how the platform could be multiplexed for downstream usage (e.g., target multiple pathways simultaneously with different strength of downregulation).

(Response 3-7) We thank the reviewer's constructive comment about the pathway optimization via TI-CRISPRi. Previous studies have shown automated pathway optimization via machine learning during the Design-Build-Test-Learn (DBTL) cycle. Machine learning could effectively bridge the gap between the Learn step and the following Design step. For example, one previous study had shown the application of machine learning to analyze the lycopene titer from random combinations of promoter-RBS of metabolic genes. The authors achieved optimal lycopene production by iterative DBTL cycle. Our tunable TI-CRISPRi could be directed to fine-tune the expression level of endogenous genes by assembling each spacer with diverse mutant DR sequences as if combining diverse promoters and RBSs to those genes. This could diversify the endogenous metabolic flux toward the production of target metabolite, and the resulting titer could be automatically analyzed via machine learning to further suggest the scope of investigation in the next DBTL cycle and optimize the endogenous metabolic flux.

HamediRad, M. et al. Towards a fully automated algorithm driven platform for biosystems design. *Nat. Commun.* 10, 5150 (2019).

This is described in the revised manuscript as follows: *“Also, targeting multiple pathways with different down-regulation strengths by assembling multiple guide RNAs with diverse mutant DR sequences could be applied to diversifying the expression of genetic circuits and metabolic flux. Analyzing the resulting bioproduction via machine learning could suggest further design rules to guide RNAs and accelerate automated genetic engineering to approach the optimum of the genetic circuit and metabolic flux^{69,70}.”* (Page 18, 374-379 in the revised version)

(Comment 3-8) Lines 408-409: How were these spacers selected? What tools and criteria were used? How were off-target effects avoided? These must be addressed here and, briefly, in the main text. The authors didn't describe the potential for off-target effects anywhere in the text and this should most definitely be discussed somewhere.

(Response 3-8) We are sorry for the insufficient description of designing the spacers. In our experiment, directly targeting the Shine-Dalgarno sequence and the start codon of the target mRNA yielded robust gene knockdown. Therefore, we designed the spacer to include the translation initiation region (from the Shine-Dalgarno sequence to the start codon). Also, since the RNA-RNA hybridization is strongly affected by its own secondary structure, we rationally designed the spacer to form the least secondary structure inside itself by inspecting the RNA structure with the NUPACK website. We added this description in the materials and methods section.

For the off-target effects, we observed in Figure 1d and 1e that the guide RNAs specifically down-regulated each reporter gene without perturbing the expression of other reporters. We admit that the analysis of the genome-wide off-target effect that occurred by the TI-CRISPRi is absent due to the difficulties in performing proteome analysis to investigate the genome-wide translational fluctuations. We think this is one of the limitations of this study but also an interesting point of investigation in further studies. We added this limitation and further improvement of this study in the discussion part to give insight to the people who read this article. We greatly thank the reviewer's constructive comment.

This is described in the revised manuscript as follows:

“For the effective targeting of dCas13, the spacer sequences were rationally selected to form the least secondary structure inside itself while targeting the translation initiation region spanning from the Shine-Dalgarno sequence to the start codon of a target gene.” (Page 22, line 451-454 in the revised version)

“We observed that only the reporter genes targeted by the guide RNAs were specifically knocked down in multiplexed gene targeting, which suggests the high specificity of our system. However, due to the difficulties in performing proteome analysis, we could not inspect the genome-wide off-target perturbations of translational inhibition under the application of TI-CRISPRi. Further investigation on proteome-level perturbation caused by TI-CRISPRi could suggest a detailed understanding of the off-target effect and propose design rules of guide RNA to enhance specificity.” (Page 16-17, line 349-355 in the revised version)

(Comment 3-9) Line 36: Rephrase “better” is a weak descriptor

(Response 3-9) We appreciate the reviewer's comment. The sentence was modified as follows:

“Developing synthetic gene expression modulators such as CRISPR-Cas system and synthetic sRNA enabled high-throughput screening of genes to systematically understand the microbial system and engineer the metabolic pathways of microorganisms¹⁻⁵.” (Page 3, line 34-36 in the revised version)

(Comment 3-10) Line 38: Remove “the” before gene expression

(Response 3-10) We apologize for the mistake. The “the” was deleted in that sentence.

(Comment 3-11) Line 79: “plenty” is a weak descriptor

(Response 3-11) We appreciate the reviewer's comment. We changed the word “plenty of” into “hundreds of” based on a new reference article.

Hu, Y. et al. Metagenomic discovery of novel CRISPR-Cas13 systems. *Cell Discov.* 8, 107 (2022).

This is described in the revised manuscript as follows:

“Several previous studies have discovered hundreds of Cas13 orthologs that can bind to target RNAs across a wide range of bacterial species with the help of genome mining and classified into four subtypes (Cas13a, Cas13b, Cas13c, and Cas13d) depending on the existence of accessory proteins or the primary amino acid sequences of the Cas13 effectors¹⁴⁻¹⁷.” (Page 5, line 79-82 in the revised version)

(Comment 3-12) Lines 330-332: Condense: “... bacterial cells concealed in Tx-CRISPRi-based screening”

(Response 3-12) We appreciate the reviewer’s comment. According to this comment, the sentence was modified as follows:

“We anticipate that additional genome-wide TI-CRISPRi screening could reveal unprecedented information about the genotype-phenotype relationships in bacterial cells not discovered in the Tx-CRISPRi screening.” (Page 17, line 360-363 in the revised version)

Reviewers' Comments:

Reviewer #1:

Remarks to the Author:

My concerns have been well addressed and i support to publish manuscript in Nature Communications.

Reviewer #2:

Remarks to the Author:

The authors have addressed all my comments well and improved the manuscript with new experiments and figures.

I also see that the authors have included a data availability statement but I dont see any referencet o strain/plasmid availability and I was wondering if the authors could clarify if the plasmid/strains are available to use by other researchers (or if they are IP protected) and if they will be shared upon request or shared in a repository (e.g. addgene in case of the plasmids).

Reviewer #3:

Remarks to the Author:

Kim et al. have provided a robust response to the comments of this and other reviewers. In particular, the authors provided compelling new experimental evidence to support their claims for an improved manuscript both in terms of clarity and scientific merit. This is especially true regarding the addition of Figure 5D (part 2) and the slew of Supplementary Figures supporting their textual claims.

As suggested by Review 2 and alluded to by this reviewer (Comment 3-7), the work could further benefit from some sort of algorithm/machine learning model that predicts expression levels so that future users of TI-CRISPRi can better tune gene expression. However, such a model would likely involve a more robust computational analysis given the poor correlation from Supplementary Figure 8 and is probably beyond the scope of the work.

The manuscript is recommended for publication in Nature Communications.

Response to Reviewers' Comments

Reviewer #1

My concerns have been well addressed and i support to publish manuscript in Nature Communications.

(Response)

Thank you for giving us fruitful comments. Measuring the growth rate helped us to consider more deeply about the growth phenotype of each strains with TI-CRISPRi, especially in terms of balancing bacterial growth and 3-HP production while adopting tunable TI-CRISPRi strategy. We are excited to publish our revised manuscript with the aid of your thoughtful review in Nature Communications.

Reviewer #2

The authors have addressed all my comments well and improved the manuscript with new experiments and figures.

I also see that the authors have included a data availability statement but I don't see any reference to strain/plasmid availability and I was wondering if the authors could clarify if the plasmid/strains are available to use by other researchers (or if they are IP protected) and if they will be shared upon request or shared in a repository (e.g. Addgene in case of the plasmids).

(Response)

Thank you for giving us constructive comments to make our research more fruitful. Your comment led us to fortify our based on the additional experiments and provide our detailed opinions about the limitations and prospects of this study.

We also thank to your suggestion about availability of bacterial strains and plasmids constructed in this study. Unfortunately, since some of them are under patent application and examination, we feel sorry that uploading all of the information of these strains and plasmids in the public repositories are quite cautious. Instead, we hope to supply them to whom these are needed under request to the corresponding author. This statement was now added in the Data availability section in the manuscript.

Again, we thank to your all comments and are excited to publish our revised manuscript with the aid of your thoughtful review in Nature Communications.

Reviewer #3

Kim et al. have provided a robust response to the comments of this and other reviewers. In particular, the authors provided compelling new experimental evidence to support their claims for an improved manuscript both in terms of clarity and scientific merit. This is especially true regarding the addition of Figure 5D (part 2) and the slew of Supplementary Figures supporting their textual claims.

As suggested by Review 2 and alluded to by this reviewer (Comment 3-7), the work could further benefit from some sort of algorithm/machine learning model that predicts expression levels so that future users of TI-CRISPRi can better tune gene expression. However, such a model would likely involve a more robust computational analysis given the poor correlation from Supplementary Figure 8 and is probably beyond the scope of the work.

The manuscript is recommended for publication in Nature Communications.

(Response)

Thank you for giving us detailed and constructive comments to make our research more solid. Adopting tunable TI-CRISPRi, which was one of the main purpose of this study, were greatly enhanced by testing more guide RNA candidates following your comments. All the other comments including discussion about machine learning application and revision of certain wordings were also very helpful to us. We are excited to publish our revised manuscript with the aid of your thoughtful review in Nature Communications.